# RELATIVE POSITIONAL ENCODING FAMILY VIA UNITARY TRANSFORMATION

## ABSTRACT

Relative positional encoding is widely used in vanilla and linear transformers to represent positional information. However, existing encoding methods of a vanilla transformer are not always directly applicable to a linear transformer, because the latter requires a decomposition of the query and key representations into separate kernel functions. Nevertheless, principles to design encoding methods suitable for linear transformers remain under-studied. In this work, we put together a variety of existing encoding approaches under a canonical form and further propose a family of relative positional encoding algorithms via *unitary transformation*. Our formulation leads to a principled framework that can be used to develop new relative positional encoding methods that preserve the linear space-time complexity. Equipped with different parameters, the proposed linearized relative positional encoding (LRPE) family derives effective encoding for various applications. Experiments show that compared with existing methods, LRPE achieves competitive performance on language modeling and various challenging downstream tasks, *e.g.*, machine translation and text classification. In the meantime, it highlights a general paradigm to design broadly more relative positional encoding methods, applicable inclusively to linear and vanilla transformers.

## 1 INTRODUCTION

Transformers have achieved remarkable progress in natural language processing (Devlin et al., 2019; Radford et al., 2019; Brown et al., 2020), computer vision (Dosovitskiy et al., 2020; Liu et al., 2021; Arnab et al., 2021) and audio processing (Gulati et al., 2020). As an important ingredient in transformers, positional encoding assigns a unique representation for each position of a token in a sequence so that the transformers can sense the position of input tokens. Among these encoding methods, absolute positional encoding (Vaswani et al., 2017; Sukhbaatar et al., 2015; Devlin et al., 2019; Liu et al., 2020) maps each individual position index into a continuous encoding. Whereas relative positional encoding (Shaw et al., 2018; Su et al., 2021; Horn et al., 2021; Liutkus et al., 2021; Huang et al., 2020; Raffel et al., 2019) generates encoding for each query-key pair, representing their relative positional offset. We focus on relative positional encoding as they are not constrained by input lengths (Chen, 2021) while showing superior performance (Shaw et al., 2018).

Linear transformers Chen (2021); Qin et al. (2022); Su et al. (2021) attract more attention recently as they can achieve linear space-time complexity with respect to input sequence length, while maintaining comparable performance with vanilla transformers. Most existing linear transformers use absolute positional encoding methods to encode positional information, since most existing relative positional encoding methods are designed for vanilla transformers and are not directly applicable to linear transformers. The main cause behind this limitation is that linear transformers decompose key and value representations in the self-attention modules into separate kernel functions to achieve linear space-time complexity. Such an additional requirement on the decomposibility is not always satisfied by existing relative positional encoding methods. On the other hand, despite some individual works (Qin et al., 2022; Chen, 2021), general principles to design relative positional encoding for linear transformers remain largely under-studied. A recent work, RoPE Su et al. (2021) proposes a new set of multiplicative encoding solutions based on rotate positional encoding and can be applied to linear transformers. In Section C.7, we show that RoPE can be seen as a special form of LRPE.

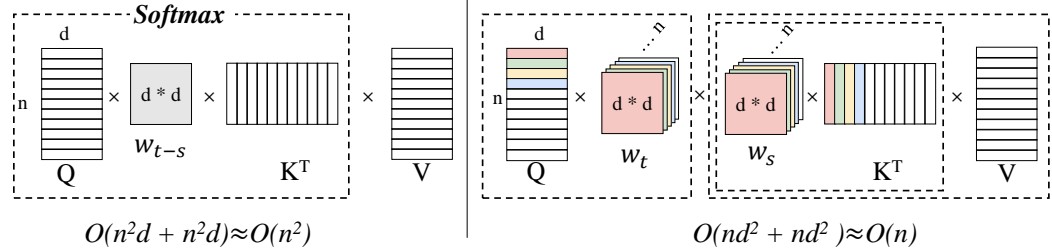

$$O(n^2d + n^2d) \approx O(n^2) \qquad\qquad O(nd^2 + nd^2) \approx O(n)$$

Figure 1: Illustration of existing relative positional encoding (left) and the proposed LRPE (right). $\mathbf{Q}$, $\mathbf{K}$, and $\mathbf{V}$ are all in the shape of $n$ by $d$, where $n$ is input length and $d$ is feature dimension. Tensors in the same dashed line box are associated for computation. In the vanilla relative positional encoding, query key attention has to be calculated first, leading to a quadratic complexity. $W_{t-s}$ refers to relative positional encoding, where $t, s$ are two positional indices on the query and key, respectively. Our LRPE achieves a decomposable encoding, *i.e.*, $W_t$ and $W_s$ are only dependent on positions of the query and key, making it fully compatible with linear transformers. When dealing with long sequences, $d \ll n$, the computation complexity is dominated by $n$, rendering $d$ negligible.

In this work, we aim to bridge this gap and study principal framework to develop relative positional encoding applicable for both linear and vanilla transformers. To this end, we start by presenting a canonical form of relative positional encoding, which reveals that differences in existing encoding methods boil down to choices of a set of query, key and relative positional matrix *primitives*. By properly selecting and composing these primitives, we could derive various existing encoding methods for vanilla (Vaswani et al., 2017; Huang et al., 2020; Shaw et al., 2018) and linear (Qin et al., 2022) transformers.

Taking advantage of the canonical form, we introduce the main contribution of our work, *i.e.*, a special family of relative positional encoding methods called *linearized relative positional encoding* (LRPE). Specifically, we supply a sufficient condition to design compatible encoding methods specially for linear transformers and prove that the linearized relative positional encoding is unitary transformation. Benefits of using unitary transformation are two-fold. On one side, since it is derived from the decomposable positional matrix, it can maintain the linear space-time complexity as shown in Fig. 1. Second, the property of the unitary transformation allows us to effectively derive the family of closed-form solutions. In particular, we show that a number of encoding methods pertain to the LRPE family, including those used in RoPE (Su et al., 2021) and PermuteFormer (Chen, 2021).

Furthermore, LRPE sheds light on a simple yet flexible theoretical paradigm to develop new effective relative positional encodings. To demonstrate this, we derive non-exhaustively three additional LRPE encoding methods by parameterizing the generic solution differently, including solutions living in either real or complex domains. Since unitary transformations are special cases of relative positional matrix, LRPE are applicable in both linear and vanilla transformers, and exclusively suitable within encoder and/or decoder layers. We experimentally demonstrate the effectiveness of the LRPE family on autoregressive and bidirectional language modelling, and on challenging downstream tasks, including machine translation and text classification. Results show that LRPE achieves competitive capability in representing relative positional information, commonly resulting in superior performance than previous encoding methods.

In summary, our main contributions are three-fold:

- We present a canonical form of relative positional encoding, which derives most existing relative positional encoding methods as its special case, including those used in linear and vanilla transformers.
- Based on the canonical form, we propose linearized relative position encoding (LRPE), a simple yet principal formulation to derive an encoding *family* that respect the linear space-time complexity in linear transformers, while being also applicable to vanilla transformers. We show several existing relative positional encoding methods in linear transformers are in LRPE family. We also provide additional particular solutions from this generic form.
- Experiments on various downstream tasks, including language modeling, machine translation and text classification show that the LRPE family show more *robust* and commonly superior results across tasks than previous relative encoding methods, are *flexible* in be-

ing plugged into linear/vanilla models, in encoder and/or decoder layers. In addition, it is *generic* to derive existing and potentially new encoding methods.

## 2 BACKGROUND AND PRELIMINARY

In this section, we provide preliminary knowledge and describe related work to facilitate the rest discussions. In the following, we denote the $k$-th row of matrix $\mathbf{M}$ as $\mathbf{m}_k^\mathsf{T}$, the $d$-dimensional identity matrix as $\mathbf{I}_d$. We omit the subscript $d$ when it is unambiguous from the context. The complete list of notations can be found in Appendix A.

### 2.1 TRANSFORMER AND ITS LINEARIZATION

We first briefly review vanilla transformer (Vaswani et al., 2017) and its linearization (Katharopoulos et al., 2020). The key component of transformer models is the self-attention block, which involves three matrices $\mathbf{Q}$ (**Query**), $\mathbf{K}$ (**Key**) and $\mathbf{V}$(**Value**); each of them is a linear projection taking $\mathbf{X} \in \mathbb{R}^{n \times d}$ as input:

$$\mathbf{Q} = \mathbf{X}\mathbf{W}_Q, \mathbf{K} = \mathbf{X}\mathbf{W}_K, \mathbf{V} = \mathbf{X}\mathbf{W}_V \in \mathbb{R}^{n \times d}. \tag{1}$$

The output $\mathbf{O} \in \mathbb{R}^{n \times d}$ is computed using the Softmax weighted sum:

$$\mathbf{O} = \text{Softmax}(\mathbf{Q}\mathbf{K}^\mathsf{T}/\sqrt{d})\mathbf{V}. \tag{2}$$

The computation overhead of the vanilla transformer grows quadratically with respect to the sequence length $n$, which becomes the bottleneck for transformers to handle long input sequences. **Linearization** of self-attention aims to reduce the computation complexity to linear (Katharopoulos et al., 2020; Ke et al., 2021; Qin et al., 2022; Vyas et al., 2020; Peng et al., 2021; Xiong et al., 2021), typically achieved via a decomposable kernel function $\phi : \mathbb{R}^d \to \mathbb{R}^{\bar{d}}$. Specifically, the output of linear attention is computed as:

$$\begin{aligned} \mathbf{O} &= \boldsymbol{\Delta}^{-1}\phi(\mathbf{Q})[\phi(\mathbf{K})^\mathsf{T}\mathbf{V}], \\ \boldsymbol{\Delta} &= \text{diag}(\phi(\mathbf{Q})[\phi(\mathbf{K})^\mathsf{T}\mathbf{1}_n]). \end{aligned} \tag{3}$$

The key property of linear attention is the **decomposability** of the kernel function. This enables to compute $\phi(\mathbf{K})^\mathsf{T}\mathbf{V} \in \mathbb{R}^{d \times d}$ first, which leads to the $O(nd^2)$ complexity, further reducing to $O(n)$ with longer inputs ($d \ll n$). See Appendix B for a detailed discussion.

### 2.2 POSITIONAL ENCODING

Self-attention is capable of parallel sequence processing but cannot capture positional information of each token. To address this issue, positional encoding methods are proposed, which can be generally categorized into two groups: absolute positional encoding and relative positional encoding.

**Absolute positional encoding** employs handcraft functions (Vaswani et al., 2017; Sukhbaatar et al., 2015) or learnable encoding lookup tables $\mathbf{P} \in \mathbb{R}^{n \times d}$ (Devlin et al., 2019; Liu et al., 2020) to represent position indices as encodings. These encodings are then combined with the context vector additively:

$$\mathbf{q}_s = \mathbf{W}_Q(\mathbf{x}_s + \mathbf{p}_s), \mathbf{k}_s = \mathbf{W}_K(\mathbf{x}_s + \mathbf{p}_s), \mathbf{v}_s = \mathbf{W}_V(\mathbf{x}_s + \mathbf{p}_s), \tag{4}$$

where the encoding formulation only depends on the absolute position index $s$, and the positional encoding size is restricted by the input sequence length.

**Relative positional encoding** considers relative position offsets between two input tokens (Shaw et al., 2018), *i.e.*,

$$\mathbf{e}_{st} = \mathbf{x}_s^\mathsf{T}\mathbf{W}_Q^\mathsf{T}\mathbf{W}_K\mathbf{x}_t + f(\mathbf{x}_s, \mathbf{x}_t, t - s), \tag{5}$$

where $s, t$ are the two positional indexes, $\mathbf{e}_{st}$ denotes the attention score before softmax. Compared to absolute positional encoding, relative positional encoding generally achieves better performance as it can handle variable input length (Chen, 2021). However, extra cost on computation and memory makes it not so efficient than absolute positional encoding (Likhomanenko et al., 2021).

Most existing relative positional encoding methods (Raffel et al., 2019; Shaw et al., 2018; Huang et al., 2020) require computing query-key attention $\mathbf{Q}\mathbf{K}^\mathsf{T}$ and combine with relative positional information, which incurs quadratic complexity. In contrast, linear attention avoids such a query-key product to achieve the linear complexity. Therefore, common relative positional encoding methods are usually not applicable in linear transformers.

## 3 OUR METHOD

In this section, we present our main technical contribution on linearized relative positional encoding, which is an encoding family that preserve the linear space-time complexity. Specifically, we start by presenting a canonical form of relative positional encoding, and show that existing encoding methods can be derived by instantiating the canonical form with different choices of so-called primitive queries, keys and positional matrices in Section 3.1. When imposing the decomposability constraint on this canonical form, we obtain a sufficient condition for linearized relative positional encoding (LRPE) and derive a family of concrete solutions in real and complex domains in Section 3.2. We provide an implementation sketch in Section 3.3.

### 3.1 CANONICAL FORM OF RELATIVE POSITIONAL ENCODING

In order to better establish connections between existing relative positional encoding methods and understand their design principles, in this section, we first present a canonical form of relative positional encoding. In particular, given a query $\mathbf{q}_s$ and key $\mathbf{k}_s$ pair, their relative positional encoding $f_{\mathrm{rel}} : \mathbb{C}^d \times \mathbb{C}^d \to \mathbb{C}$ can be represented as:

$$f_{\mathrm{rel}}(\mathbf{q}_s, \mathbf{k}_t) = \sum_{l=1}^{m} (\hat{\mathbf{q}}_s^{(l)})^{\mathsf{H}} \mathbf{W}_{t-s}^{(l)} \hat{\mathbf{k}}_t^{(l)}, \tag{6}$$

where $\mathsf{H}$ represents **conjugate transposition** and $m$ represents number of primitives. We refer $\hat{\mathbf{q}}_s^{(l)} \in \mathbb{C}^{d_1^{(l)}}, \hat{\mathbf{k}}_t^{(l)} \in \mathbb{C}^{d_2^{(l)}}, \mathbf{W}_{t-s}^{(l)} \in \mathbb{C}^{d_1^{(l)} \times d_2^{(l)}}$ as query, key and relative positional matrix *primitives*, respectively, used as constituent components to construct the relative positional encoding. Note that query primitives do not always indicate a reliance on query embeddings, similarly for other primitives. For example, an identify matrix can also serve as primitives, as we will show shortly in Section 3.1.1.

To demonstrate Eq. 6 is a generic formulation, we show that it flexibly induces a wide range of existing relative encoding methods (Shaw et al., 2018; Su et al., 2021; Horn et al., 2021; Liutkus et al., 2021; Huang et al., 2020; Raffel et al., 2019) by selecting and compositing different choices of primitives. Among them, we highlight two examples in the following section, and leave the complete discussions in the Appendix C.1.

#### 3.1.1 TYPICAL ENCODING EXAMPLES FROM THE CANONICAL FROM

**Additive.** In (Huang et al., 2020), the relative positional encoding is formulated as an extra additive term to the query-key inner-product:

$$f_{\mathrm{rel}}(\mathbf{q}_s, \mathbf{k}_t) = \mathbf{q}_s^{\mathsf{H}} \mathbf{k}_t + w_{t-s}, \tag{7}$$

which can be derived by including an extra identity term as a primitive, formally denoted as:

$$m = 2,$$
$$\hat{\mathbf{q}}_s^{(1)} = \mathbf{q}_s, \hat{\mathbf{k}}_t^{(1)} = \mathbf{k}_t, \mathbf{W}_{t-s}^{(1)} = \mathbf{I}_d, \tag{8}$$
$$\hat{\mathbf{q}}_s^{(2)} = \mathbf{I}_d, \hat{\mathbf{k}}_t^{(2)} = \mathbf{I}_d, \mathbf{W}_{t-s}^{(2)} = w_{t-s} \mathbf{I}_d.$$

**Multiplicative.** In RoPE (Su et al., 2021), the relative positional encoding works in the form of the weighted inner product:

$$f_{\mathrm{rel}}(\mathbf{q}_s, \mathbf{k}_t) = \mathbf{q}_s^{\mathsf{H}} \mathbf{W}_{t-s} \mathbf{k}_t, \tag{9}$$

which can be denoted as:

$$m = 1,$$
$$\hat{\mathbf{q}}_s^{(1)} = \mathbf{q}_s, \hat{\mathbf{k}}_t^{(1)} = \mathbf{k}_t, \mathbf{W}_{t-s}^{(1)} = \mathbf{W}_{t-s}. \tag{10}$$

### 3.1.2 SIMPLIFICATION

For the ease of remaining discussion, we introduce necessary notations and simplify Eq. 6.

$$\hat{d}_1 = \sum_{l=1}^{m} d_1^{(l)}, \hat{d}_2 = \sum_{l=1}^{m} d_2^{(l)},$$

$$\hat{\mathbf{q}}_s = \left[ (\hat{\mathbf{q}}_s^{(1)})^{\mathsf{T}}, \dots, (\hat{\mathbf{q}}_s^{(m)})^{\mathsf{T}} \right]^{\mathsf{T}} \in \mathbb{C}^{\hat{d}_1}, \hat{\mathbf{k}}_t = \left[ (\hat{\mathbf{k}}_t^{(1)})^{\mathsf{T}}, \dots, (\hat{\mathbf{k}}_t^{(m)})^{\mathsf{T}} \right]^{\mathsf{T}} \in \mathbb{C}^{\hat{d}_2}, \tag{11}$$

$$\hat{\mathbf{W}}_{t-s} = \text{block-diag}\{\mathbf{W}_{t-s}^{(1)} \dots, \mathbf{W}_{t-s}^{(m)}\} \in \mathbb{C}^{\hat{d}_1 \times \hat{d}_2}.$$

with these notations, we can rewrite Eq. 6 into the matrix form: $f_{\text{rel}}(\mathbf{q}_s, \mathbf{k}_t) = \hat{\mathbf{q}}_s^{\mathsf{H}} \hat{\mathbf{W}}_{t-s} \hat{\mathbf{k}}_t$. Since every component of $\hat{\mathbf{q}}_s$ and $\hat{\mathbf{k}}_t$ are handled with no difference, without losing generality, we only discuss cases where $m = 1$:

$$f_{\text{rel}}(\mathbf{q}_s, \mathbf{k}_t) = \mathbf{q}_s^{\mathsf{H}} \mathbf{W}_{t-s} \mathbf{k}_t. \tag{12}$$

## 3.2 LINEARIZED RELATIVE POSITION ENCODING

Eq. 6 is a canonical form of relative positional encoding, meaning that its variants are applicable to vanilla transformers but not necessarily for linear ones. To design relative encoding compatible with linear transformers, the attention computation has to respect the decomposibility condition. This additional condition leads to the linearized relative position encoding (LRPE) family, defined as follows.

**Definition 3.1.** *A relative position encoding is called linearized relative position encoding (*LRPE*), when the following holds:*

$$\forall \mathbf{q}_s, \mathbf{k}_t \in \mathbb{C}^d, f_{\text{rel}}(\mathbf{q}_s, \mathbf{k}_t) = \mathbf{q}_s^{\mathsf{H}} \mathbf{W}_{t-s} \mathbf{k}_t = (\mathbf{M}_s \mathbf{q}_s)^{\mathsf{H}} (\mathbf{M}_t \mathbf{k}_t) = \mathbf{q}_s^{\mathsf{H}} \mathbf{M}_s^{\mathsf{H}} \mathbf{M}_t \mathbf{k}_t, \tag{13}$$

*where* $\mathbf{q}_s, \mathbf{k}_t \in \mathbb{C}^d$, $\mathbf{W}_s, \mathbf{M}_s \in \mathbb{C}^{d \times d}$, $\mathbf{W}_0 = \mathbf{I}_d$.

The assumption of $\mathbf{W}_0 = \mathbf{I}_d$ implies that the interaction between tokens from the same position only depends on the content, which is reasonable enough that most encoding methods respect. In its essence, Eq. 13 ensures the positional matrix is decomposable. In this way, the query-key inner-product can be avoided in the attention computation. Consequently, complexity of computing LRPE is $O(nd^2)$, where $n$ is sequence length, $d$ is embedding dimension as Appendix C.2 shows in detail.

We prove that Eq. 13 can be simplified based on the following proposition:

**Proposition 3.2.** *Eq. 13 is equivalent to Eq. 14 and* $\mathbf{W}_t$ *is Unitary matrix,*

$$\mathbf{W}_{t-s} = \mathbf{W}_s^{\mathsf{H}} \mathbf{W}_t. \tag{14}$$

*Proof of Proposition 3.2.* According to the arbitrariness of $\mathbf{q}_s, \mathbf{k}_t$, Eq. 13 is equivalent to

$$\mathbf{W}_{t-s} = \mathbf{M}_s^{\mathsf{H}} \mathbf{M}_t. \tag{15}$$

Take $s = t$ in Eq 13, we get (since we assume that $\mathbf{W}_0 = \mathbf{I}_d$):

$$\mathbf{M}_s^{\mathsf{H}} \mathbf{M}_s = \mathbf{W}_0 = \mathbf{I}_d. \tag{16}$$

Thus, $\mathbf{M}_s$ is a unitary matrix. On the other hand, note that for any unitary matrix $\mathbf{P}$, we always have

$$\mathbf{W}_{t-s} = \mathbf{M}_s^{\mathsf{H}} \mathbf{M}_t = \mathbf{M}_s^{\mathsf{H}} \mathbf{I}_d \mathbf{M}_t = \mathbf{M}_s^{\mathsf{H}} \mathbf{P}^{\mathsf{H}} \mathbf{P} \mathbf{M}_t = (\mathbf{P} \mathbf{M}_s)^{\mathsf{H}} (\mathbf{P} \mathbf{M}_t). \tag{17}$$

This means that left multiplying $\mathbf{M}_t$ by a unitary matrix $\mathbf{P}$ does not change Eq. 13. Since $\mathbf{M}_s$ and $\mathbf{M}_0^{\mathsf{H}}$ are also unitary matrices, we can perform the following transformation:

$$\overline{\mathbf{M}}_s = \mathbf{M}_0^{\mathsf{H}} \mathbf{M}_s. \tag{18}$$

With $\overline{\mathbf{M}}_s$, Eq. 15 becomes

$$\mathbf{W}_{t-s} = \overline{\mathbf{M}}_s^{\mathsf{H}} \overline{\mathbf{M}}_t. \tag{19}$$

Take $s = 0$, we have

$$\mathbf{W}_t = \overline{\mathbf{M}}_0^{\mathsf{H}} \overline{\mathbf{M}}_t = \mathbf{M}_0^{\mathsf{H}} \mathbf{M}_0 \overline{\mathbf{M}}_t = \mathbf{I}_d \overline{\mathbf{M}}_t = \overline{\mathbf{M}}_t. \tag{20}$$

Thus Eq. 19 becomes

$$\mathbf{W}_{t-s} = \mathbf{W}_s^{\mathsf{H}} \mathbf{W}_t. \tag{21}$$

Since $\overline{\mathbf{M}}_s$ is a unitary matrix, $\mathbf{W}_s$ is also a unitary matrix, *i.e.*,

$$\mathbf{W}_s^{\mathsf{H}} \mathbf{W}_s = \mathbf{I}_d. \tag{22}$$

$\square$

The detailed proof can be found in Appendix 3.2. In the following section, we derive some particular solutions for Eq. 14.

### 3.2.1 PARTICULAR SOLUTIONS

In this section, we discuss Eq. 14 and give a family of solutions. It is worth noting that the solutions we provide are all in the form of $\mathbf{W}_s = \mathbf{P}^{\mathsf{H}} \mathbf{\Lambda}^{(s)} \mathbf{P}$, where $\mathbf{P}, \mathbf{\Lambda}^{(s)}$ are unitary matrix. The complete derivation can be found in Appendix C.4, C.5, C.6.

Table 1: LRPE variants. The rows of the table represent the type of the $\mathbf{P}$ matrix, and the columns represent the type of the $\mathbf{\Lambda}^{(s)}$ matrix.

| $\mathbf{\Lambda}^{(s)}$ type  $\mathbf{P}$ type | Unitary (Solution 1) | Orthogonal (Solution 2) | Orthogonal learnable (Solution 2) | Permutation (Solution 3) |
|---|---|---|---|---|
| Householder | - | Type1 | Type2 | Type4 |
| Householder learnable | - | - | Type3 | - |
| Permutation | - | Type5 | Type6 | Type7 |
| FFT | Type8 | - | - | - |

**Unitary (Solution 1)**   The first case is discussed in the complex domain, which is not common in transformer models yet exhibiting an elegant solution.

**Proposition 3.1.** *The following form of $\mathbf{W}_s \in \mathbb{C}^{d \times d}$ satisfies Eq. 14:*

$$\begin{aligned}
\mathbf{W}_s &= \mathbf{P}^{\mathsf{H}} \mathbf{\Lambda}^{(s)} \mathbf{P}, \\
\mathbf{\Lambda}^{(s)} &= \mathrm{diag}\{\exp(is\alpha_1), \ldots, \exp(is\alpha_d)\},
\end{aligned} \tag{23}$$

*where $\mathbf{P} \in \mathbb{C}^{d \times d}$ is **unitary** matrix, $\alpha_k, k = 1, \ldots, d$ are parameters.*

**Orthogonal (Solution 2)**   Now we consider the real domain, a more general case in transformers.

**Proposition 3.2.** *The following form of $\mathbf{W}_s \in \mathbb{R}^{d \times d}$ satisfies Eq. 14:*

$$\mathbf{W}_s = \mathbf{P}^{\mathsf{T}} \mathbf{\Lambda}^{(s)} \mathbf{P}, \mathbf{\Lambda}^{(s)} = \begin{bmatrix} \mathbf{A}^{(s)} & \\ & \mathbf{B}^{(s)} \end{bmatrix},$$

$$\mathbf{A}^{(s)} = \begin{bmatrix} \mathbf{A}_1^{(s)} & & \\ & \ddots & \\ & & \mathbf{A}_p^{(s)} \end{bmatrix} \in \mathbb{R}^{2p \times 2p}, \mathbf{B}^{(s)} = \mathbf{I}_q \in \mathbb{R}^{q \times q}, \mathbf{A}_k^{(s)} = \begin{bmatrix} \cos(s\alpha_k) & -\sin(s\alpha_k) \\ \sin(s\alpha_k) & \cos(s\alpha_k) \end{bmatrix},$$

$$\tag{24}$$

*where $\mathbf{P} \in \mathbb{R}^{d \times d}$ is **orthogonal** matrix, $\alpha_k, k = 1, \ldots, d$ are parameters.*

**Permutation (Solution 3)**   The last case is inspired by PermuteFormer (Chen, 2021), which is associated with the permutation matrix:

**Proposition 3.3.** *The following form of $\mathbf{W}_k \in \mathbb{R}^{d \times d}$ satisfies Eq. 14:*

$$\begin{aligned}
\mathbf{W}_k &= \mathbf{P}^{\mathsf{T}} \mathbf{\Lambda}^{(k)} \mathbf{P}, \\
\pi : \{1, 2, \cdots, d\} &\to \{1, 2, \cdots, d\} \text{ is permutation}, \\
\mathbf{\Lambda}^{(k)} &= (\mathbf{I})_{\pi^k},
\end{aligned} \tag{25}$$

*where $\mathbf{P} \in \mathbb{R}^{d \times d}$ is the **orthogonal** matrix.*

## 3.3 THE LRPE FAMILY

LRPE ($\mathbf{W}_s = \mathbf{P}^H \mathbf{\Lambda}^{(s)} \mathbf{P}$) contains two components, *i.e.*, a fixed unitary matrix $\mathbf{P}$ and a unitary matrix family $\mathbf{\Lambda}^{(s)}$ as mentioned in proposition 3.1, 3.2, and 3.3. The $\mathbf{P}$ can be seen as a rotation matrix that rotates the token feature to a particular coordinate system and the $\mathbf{\Lambda}^{(s)}$ derives the positional information from the rotated feature.

In this paper, we select three types of commonly used orthogonal matrix, *i.e.*, (1) householder matrix (Golub & Van Loan, 2013), (2) permutation matrix, and (3) FFT matrix (Bracewell & Bracewell, 1986). We combine the selected Matrix $\mathbf{P}$ with three LRPE solutions above to obtain 8 practical LRPE types as shown in Table 1. Detailed information can be found in Appendix C.7.

## 4 EXPERIMENTS

Table 2: Quantitative results of the autoregressive language model on the WikiText-103 dataset. The best result is highlighted with **bold** and the second best with underlined. ↓ means *smaller is better*.

| Method | | Linear attention | | Vanilla attention | |
|---|---|---|---|---|---|
| | | PPL (val)↓ | PPL (test)↓ | PPL (val)↓ | PPL (test)↓ |
| **Competitors** | Base | 33.94 | 33.74 | 30.82 | 29.78 |
| | RoPE | 33.40 | 33.13 | 30.13 | 29.31 |
| | SPE | 43.50 | 41.91 | 32.99 | 32.26 |
| | PER | 32.86 | **32.53** | 32.21 | 31.95 |
| | T5 | - | - | 30.70 | 29.74 |
| | XL | - | - | 32.93 | 34.36 |
| **Householder** | Type1 | 33.65 | 33.58 | 29.66 | 28.80 |
| | Type2 | **32.83** | 32.80 | **29.60** | **28.69** |
| | Type3 | 33.04 | 32.81 | 29.62 | 28.81 |
| | Type4 | 35.05 | 34.80 | 31.66 | 31.32 |
| **Permutation** | Type5 | 34.24 | 33.89 | 29.92 | 29.10 |
| | Type6 | 33.13 | 33.05 | 29.83 | 28.87 |
| | Type7 | 34.07 | 33.87 | 31.96 | 31.37 |
| **FFT** | Type8 | 36.78 | 36.31 | 29.85 | 28.96 |

Figure 2: Validation PPL of linear (left) and vanilla attention (right) of the bidirectional language model pretrained on the WikiText-103 dataset. In both cases, the best result of proposed LRPE has a better PPL and faster converge speed than competing methods.

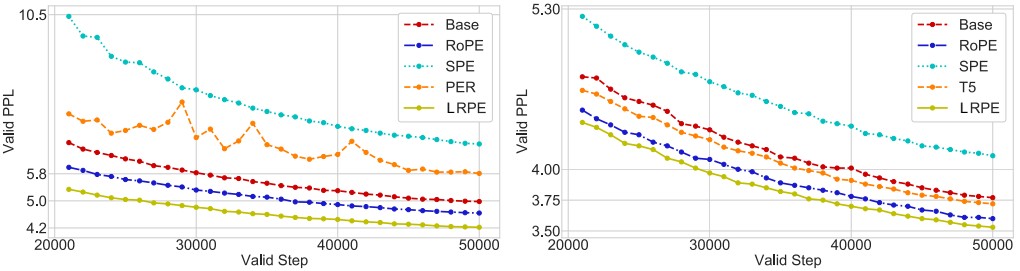

## 4.1 EXPERIMENTAL SETTINGS

**Primary tasks.** We validate the effectiveness of the proposed LRPE on various NLP tasks that resort to different Transformer architectures. Specifically, we first study the autoregressive language model (Radford et al., 2018) with a GPT-like decoder-only structure. This is followed by the bidirectional language model (*encoder-only*), which adopts the Roberta architecture (Liu et al., 2020) and is pretrained and then fine-tuned on several downstream tasks from the GLUE benchmark (Wang et al., 2018). Then, we evaluate LRPE on machine translation (*encoder-decoder*).

**Competing methods.** Our baseline is the Transformer model (Vaswani et al., 2017) without relative positional encoding. For comparison, we also choose four state-of-the-art methods , *i.e.*, RoPE (Su et al., 2021), SPE (Liutkus et al., 2021), PermutateFormer (abbreviated as "PER") (Chen, 2021), T5 (Raffel et al., 2019) and Transformer-XL (Dai et al., 2019) (abbreviated as "XL"), and test them in

both linear attention and vanilla attention. In the linear attention, we employ $1+\text{elu}(\cdot)$ (Katharopoulos et al., 2020) as the kernel function.

**Training configuration.** Our experiments are implemented in the *Fairseq* framework (Ott et al., 2019) with V100 GPUs. All the methods share the same configurations which are listed in Appendix D.1.

Table 3: Quantitative results of the Roberta model fine-tuned on the GLUE dataset. MNLI is reported by the match/mismatch splits. All the downstream tasks are measured by the accuracy. The best result is highlighted with **bold** and the second with underlined. ↑ means *larger is better*.

| Attention | Method | MNLI | QNLI | QQP | SST-2 | AVG↑ |
|---|---|---|---|---|---|---|
| **Linear** | Base | **74.87/ 75.37** | 82.59 | **88.17** | 87.27 | 81.65 |
| | RoPE | 67.13/67.69 | 79.97 | 85.11 | 78.21 | 77.61 |
| | SPE | 67.07/68.51 | 74.79 | 71.43 | 78.67 | 72.99 |
| | PER | 67.43/69.26 | 76.13 | 83.93 | 86.93 | 78.61 |
| | Type5 | 74.66/75.05 | **83.10** | 85.98 | **88.30** | **83.01** |
| **Vanilla** | Base | **79.37**/79.07 | **87.79** | **88.33** | **90.25** | **86.44** |
| | RoPE | 75.21/76.06 | 86.20 | 87.34 | 87.73 | 84.12 |
| | SPE | 76.01/76.61 | 85.67 | 87.66 | 89.33 | 84.67 |
| | T5 | 77.83/78.84 | 86.93 | 87.78 | 89.91 | 85.61 |
| | Type1 | 79.18/**79.85** | 87.57 | 87.89 | 89.11 | 85.94 |

## 4.2 RESULTS IN LINEAR SETTING

**Autoregressive language model.** The autoregressive language model has 6 decoder layers and is trained on the WikiText-103 dataset (Merity et al., 2017). We use the Perplexity (PPL) as the evaluation metric and report the results in Table 2. We observe that under the linear setting, most variants of LRPE present performance gain over the baseline. Our best model, *i.e.*, Type2, outperforms RoPE and SPE on both validation and test sets to a large margin, and achieves comparable results to PER with minor difference. Clearly, the proposed method is effective in encoding causal data.

**Bidirectional language model.** The bidirectional model follows an encoder-only structure, *i.e.*, Roberta (Liu et al., 2020), with 12 layers. We first pretrain it on the WikiText-103 dataset, and present the results in Fig. 2 and Appendix D.2. Generally, LRPE (Type2 in this case) has better performance, *i.e.*, smaller validation PPL in all evaluation steps, than competing methods. Notably, it surpasses RoPE, SPE and PER by nearly 10%, 27% and 37% in terms of the final PPL, indicating its superiority in bidirectional language modeling.

We then fine-tune the pretrained model on the GLUE dataset (Wang et al., 2018). We use different learning rates among 1e-5, 3e-5, 6e-5, 1e-4 and choosing the best result after fine-tuning for 3 epochs. From Table 3, we find that the two representative LRPE variants, *i.e.*, *Type5*, perform consistently better than other methods in all metrics. The average score of *Type5* beats RoPE, SPE, and PER by more than 4.4%.

**Machine translation.** For this task, we adopt the base transformer model which consists of 6 encoder layers and 6 decoder layers, and train it on the WMT'14 En-De dataset (Bojar et al., 2014). We ran each experiment 5 times and report the averaged results. Note that in practical, we only embed the linear attention and its corresponding relative positional encoding in encoders, since we empirically find that the model cannot converge appropriately when the linear attention appears in decoders. We measure the accuracy with BLEU, and the quantitative results on both validation and test sets are displayed in Table 4. Most variants in the LRPE family have comparable performances to the competing methods on the validation set, and Type4 ranks first on the test data, which demonstrates again the validity of our LRPE.

However, a few variants present less competitive results than the state-of-the-arts on the test data. Empirically, this is caused by the relatively high sensitivity of parameter tuning on the machine translation performance while all listed methods share the identical parameter setting. We will concentrate on how to further improve their accuracy by specializing the parameters for each variant in the future work.

Table 4: Quantitative results of machine translation on the WMT'14 En-De dataset. Evaluation metrics include the validation loss, validation BLEU (Papineni et al., 2002), and test SACRE_BLEU (Post, 2018). The best result is highlighted with **bold** and the second with underlined. ↑ means *larger is better*. $\pm\Delta$ means standard deviation.

| Method | | Linear attention | | Vanilla attention | |
|---|---|---|---|---|---|
| | | BLEU (val)↑ | BLEU (test)↑ | BLEU (val)↑ | BLEU (test)↑ |
| **Baselines** | Base | $29.57 \pm 0.06$ | $26.34 \pm 1.05$ | $29.92 \pm 0.3$ | $27.50 \pm 0.18$ |
| | RoPE | $29.60 \pm 0.04$ | $26.32 \pm 0.36$ | $30.09 \pm 0.06$ | $27.25 \pm 0.39$ |
| | SPE | $29.39 \pm 0.02$ | $26.74 \pm 0.12$ | $29.76 \pm 0.06$ | $27.34 \pm 0.12$ |
| | PER | **$29.84 \pm 0.04$** | **$27.47 \pm 0.1$** | $29.73 \pm 0.08$ | $27.19 \pm 0.30$ |
| | T5 | - | - | $29.95 \pm 0.09$ | **$27.56 \pm 0.22$** |
| **Householder** | Type1 | $29.60 \pm 0.05$ | $26.16 \pm 0.51$ | $30.09 \pm 0.02$ | $27.18 \pm 0.27$ |
| | Type2 | $29.65 \pm 0.04$ | $26.13 \pm 0.77$ | $30.07 \pm 0.02$ | $27.28 \pm 0.37$ |
| | Type3 | $29.68 \pm 0.03$ | $26.02 \pm 0.61$ | $30.08 \pm 0.03$ | $27.44 \pm 0.16$ |
| | Type4 | $29.78 \pm 0.05$ | **$27.47 \pm 0.45$** | $29.95 \pm 0.05$ | $27.41 \pm 0.16$ |
| **Permutation** | Type5 | $29.61 \pm 0.04$ | $26.13 \pm 0.5$ | $30.08 \pm 0.05$ | $27.18 \pm 0.52$ |
| | Type6 | $29.71 \pm 0.04$ | $26.13 \pm 1.16$ | $30.09 \pm 0.02$ | $27.39 \pm 0.49$ |
| | Type7 | $29.82 \pm 0.03$ | $27.45 \pm 0.12$ | $29.96 \pm 0.03$ | $27.21 \pm 0.28$ |
| **FFT** | Type8 | - | - | **$30.10 \pm 0.06$** | $27.46 \pm 0.22$ |

Table 5: Ablation results with different rotation matrix $\mathbf{P}$ for language modeling on the WikiText-103 dataset.

| Method | Linear attention | | Vanilla attention | |
|---|---|---|---|---|
| | PPL (val)↓ | PPL (test)↓ | PPL (val)↓ | PPL (test)↓ |
| **Householder** | 32.83 | 32.80 | 29.60 | 28.69 |
| **Permutation** | 33.13 | 33.05 | 29.83 | 28.87 |
| **Identity** | 34.09 | 33.70 | 30.05 | 29.24 |

### 4.3 MODEL ANALYSIS

**An explanation of LRPE.** According to the discussion in Section. 3.3, The LRPE rotates the token feature through $\mathbf{P}$, and encodes the positional information through $\mathbf{\Lambda}^{(s)}$. In Table 5, we ablate the effectiveness of the $\mathbf{P}$ matrix on the autoregressive language modeling task. Our approach with the Householder matrix and the Permutation matrix achieve marginally better results than the one without rotation (Identity matrix). It indicates that we can get better performance by carefully selecting the projection of the positional encoding.

**Complexity and efficiency.** The implementation of the proposed LRPE does not affect the computational complexity of the linear transformer, *i.e.*, preserving the linear complexity as $O(n)$. We also measure the training speed of the bidirectional language model on the same local machine (*i.e.*, a GeForce GTX 1060 card), and observe that the speed after using LRPE is only 9% slower than the baseline on average. The detailed comparison of the efficiency can be found in Appendix D.3. In general, UPRE does not incur significant computational burden to the transformer, and can fulfill the practical needs by maintaining comparable efficiency.

**Generalization to vanilla attention.** Finally, we investigate the generalization of LRPE towards the vanilla attention. The results are reported in Fig. 2, Table 2, 3, and 4. The conclusion is consistent with that of the linear setting, *i.e.*, improving the vanilla transformer baseline and achieving competitive performance to the competing methods. It indicates the good flexibility of LRPE as it can be seamlessly applied to any attention type.

### CONCLUSION

In this paper, we standardize the form of relative positional encoding in both linear and vanilla transformers, and focus the case in the linear attention. The unitary transformation is employed as a special solution to the linearized relative positional encoding, and the solutions as per various constraints constitute the unitary relative positional encoding (LRPE) family. We validate the effectiveness of LRPE through extensive experiments on several NLP tasks with different transformer architectures. It outperforms state-of-the-art methods under both linear and vanilla settings.

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

# Appendix

## A  MATHEMATICAL NOTATIONS

| Notation | Meaning |
|---|---|
| $\mathbf{X}$ | Hidden state. |
| $\mathbf{Q}, \mathbf{K}, \mathbf{V}$ | Query, key, value. |
| $\mathbf{W}_Q, \mathbf{W}_K, \mathbf{W}_V$ | Weight matrices for $\mathbf{Q},\mathbf{K},\mathbf{V}$. |
| $\mathbf{O}$ | Attention output. |
| $\mathbf{m}_s^\mathsf{T}$ | $s$-th row of matrix $M$ (real domain). |
| $\mathbf{m}_s^\mathsf{H}$ | $s$-th row of matrix $M$ (complex domain). |
| $\phi$ | Kernel function for linear attention. |
| $\mathbf{1}_d$ | All-ones vector with dimention $d$. |
| $\mathbf{I}_d$ | Identity matrix with dimention $d$. |
| block-diag | Combining matrices into larger block diagonal matrices as in Eq. 26 |

Table 6: Mathematical notations used in the paper.

$$\text{block-diag}\{\mathbf{W}_1, \mathbf{W}_2, \ldots, \mathbf{W}_n\} = \begin{bmatrix} \mathbf{W}_1 & & & \\ & \mathbf{W}_2 & & \\ & & \ddots & \\ & & & \mathbf{W}_n \end{bmatrix}. \tag{26}$$

## B  COMPUTATION OF VANILLA/LINEAR ATTENTION

### B.1  BASIC NOTATIONS

Both vanilla and linear attention blocks involve three matrices, *i.e.*, $\mathbf{Q}$ (**Query**), $\mathbf{K}$ (**Key**) and $\mathbf{V}$ (**Value**). All of them are linear projections of input $\mathbf{X} \in \mathbb{C}^{n \times d}$, *i.e.*,

$$\mathbf{X} = \begin{bmatrix} \mathbf{x}_1^\mathsf{T} \\ \vdots \\ \mathbf{x}_n^\mathsf{T} \end{bmatrix} \in \mathbb{R}^{n \times d},$$

$$\mathbf{Q} = \begin{bmatrix} \mathbf{q}_1^\mathsf{T} \\ \vdots \\ \mathbf{q}_n^\mathsf{T} \end{bmatrix} = \mathbf{X}\mathbf{W}_Q = \begin{bmatrix} \mathbf{x}_1^\mathsf{T}\mathbf{W}_Q \\ \vdots \\ \mathbf{x}_n^\mathsf{T}\mathbf{W}_Q \end{bmatrix} \in \mathbb{R}^{n \times d},$$

$$\mathbf{K} = \begin{bmatrix} \mathbf{k}_1^\mathsf{T} \\ \vdots \\ \mathbf{k}_n^\mathsf{T} \end{bmatrix} = \mathbf{X}\mathbf{W}_K = \begin{bmatrix} \mathbf{x}_1^\mathsf{T}\mathbf{W}_K \\ \vdots \\ \mathbf{x}_n^\mathsf{T}\mathbf{W}_K \end{bmatrix} \in \mathbb{R}^{n \times d}, \tag{27}$$

$$\mathbf{V} = \begin{bmatrix} \mathbf{v}_1^\mathsf{T} \\ \vdots \\ \mathbf{v}_n^\mathsf{T} \end{bmatrix} = \mathbf{X}\mathbf{W}_V = \begin{bmatrix} \mathbf{x}_1^\mathsf{T}\mathbf{W}_V \\ \vdots \\ \mathbf{x}_n^\mathsf{T}\mathbf{W}_V \end{bmatrix} \in \mathbb{R}^{n \times d},$$

where $\mathbf{W}_Q, \mathbf{W}_K, \mathbf{W}_V \in \mathbb{R}^{d \times d}$.

The vector form is organized as

$$\mathbf{q}_s = \mathbf{W}_Q^\mathsf{T}\mathbf{x}_s, \mathbf{k}_s = \mathbf{W}_K^\mathsf{T}\mathbf{x}_s, \mathbf{v}_s = \mathbf{W}_V^\mathsf{T}\mathbf{x}_s. \tag{28}$$

The attention output is

$$\mathbf{O} = \begin{bmatrix} \mathbf{o}_1^\mathsf{T} \\ \vdots \\ \mathbf{o}_n^\mathsf{T} \end{bmatrix} \in \mathbb{R}^{n \times d}. \tag{29}$$

### B.2 VANILLA ATTENTION

In vanilla attention, the output is computed using the Softmax weighted sum, *i.e.*,

$$
\begin{aligned}
\mathbf{o}_s &= \text{Attention}(\mathbf{q}_s, \mathbf{K}, \mathbf{V}) \\
&= \sum_{t=1}^{n} \mathbf{a}_{st}\mathbf{v}_t \\
&= \sum_{t=1}^{n} \frac{\exp\left(\mathbf{q}_s^{\mathsf{T}}\mathbf{k}_t/\sqrt{d}\right)\mathbf{v}_t}{\sum_{r=1}^{n}\exp\left(\mathbf{q}_s^{\mathsf{T}}\mathbf{k}_r/\sqrt{d}\right)}, \\
\mathbf{O} &= \text{Softmax}(\mathbf{Q}\mathbf{K}^{\mathsf{T}}/\sqrt{d})\mathbf{V}.
\end{aligned}
\tag{30}
$$

### B.3 LINEAR ATTENTION

The linear attention is formulated as follows,

$$
\begin{aligned}
\mathbf{o}_s &= \text{LinearAttention}(\mathbf{q}_s, \mathbf{K}, \mathbf{V}) \\
&= \sum_{t=1}^{n} \mathbf{a}_{st}\mathbf{v}_t \\
&= \sum_{t=1}^{n} \frac{\phi(\mathbf{q}_s)^{\mathsf{T}}\phi(\mathbf{k}_t)}{\sum_{t=1}^{n}\phi(\mathbf{q}_s)^{\mathsf{T}}\phi(\mathbf{k}_t)}\mathbf{v}_t \\
&= \frac{\sum_{t=1}^{n}\phi(\mathbf{q}_s)^{\mathsf{T}}\phi(\mathbf{k}_t)\mathbf{v}_t}{\sum_{t=1}^{n}\phi(\mathbf{q}_s)^{\mathsf{T}}\phi(\mathbf{k}_t)} \\
&= \phi(\mathbf{q}_s)^{\mathsf{T}}\frac{\sum_{t=1}^{n}\phi(\mathbf{k}_t)\mathbf{v}_t}{\phi(\mathbf{q}_s)^{\mathsf{T}}\sum_{t=1}^{n}\phi(\mathbf{k}_t)}, \\
\mathbf{O} &= \mathbf{\Delta}^{-1}\phi(\mathbf{Q})\phi(\mathbf{K})^{\mathsf{T}}\mathbf{V} \\
&= \mathbf{\Delta}^{-1}\phi(\mathbf{Q})[\phi(\mathbf{K})^{\mathsf{T}}\mathbf{V}], \\
\mathbf{\Delta} &= \text{diag}(\phi(\mathbf{Q})[\phi(\mathbf{K})^{\mathsf{T}}\mathbf{1}_n]).
\end{aligned}
\tag{31}
$$

## C PROOF OF THEOREM

### C.1 MORE EXAMPLES

In the following, we provide two additional examples of relative positional encoding with the canonical form.

**RPR** (Shaw et al., 2018):

$$
\begin{aligned}
f_{\text{rel}}(\mathbf{q}_s, \mathbf{k}_t) &= \mathbf{q}_s^{\mathsf{H}}\mathbf{k}_t + \mathbf{q}_s^{\mathsf{H}}\mathbf{c}_{t-s}, \\
\mathbf{c}_{t-s} &= \mathbf{w}_{\text{clip}(t-s,k)}, \\
\text{clip}(x, k) &= \max(-k, \min(k, x)), \\
\mathbf{w_s} &\in \mathbb{C}^d, -k \le s \le k.
\end{aligned}
\tag{32}
$$

The canonical form is

$$
\begin{aligned}
m &= 2, \\
\hat{\mathbf{q}}_s^{(1)} = \mathbf{q}_s, \hat{\mathbf{k}}_t^{(1)} &= \mathbf{k}_t, \mathbf{W}_{t-s}^{(1)} = \mathbf{I}_d, \\
\hat{\mathbf{q}}_s^{(2)} = \mathbf{q}_s, \hat{\mathbf{k}}_t^{(2)} &= \mathbf{I}_d, \mathbf{W}_{t-s}^{(2)} = \frac{1}{d}\underbrace{[\mathbf{c}_{t-s} \quad \dots \quad \mathbf{c}_{t-s}]}_{d \text{ columns}}.
\end{aligned}
\tag{33}
$$

**DeBERTa** (Huang et al., 2020):

$$f_{\mathrm{rel}}(\mathbf{q}_s, \mathbf{k}_t) = \mathbf{q}_s^{\mathsf{H}}\mathbf{k}_t + \mathbf{q}_s^{\mathsf{H}}\bar{\mathbf{k}}_{g(s-t)} + \bar{\mathbf{q}}_{g(t-s)}^{\mathsf{H}}\mathbf{k}_t,$$

$$g(x) = \begin{cases} 0 & x \leq -c \\ 2c-1 & x \geq c \\ x+c & \text{others.} \end{cases} \tag{34}$$

The canonical form is

$$m = 3,$$
$$\hat{\mathbf{q}}_s^{(1)} = \mathbf{q}_s, \hat{\mathbf{k}}_t^{(1)} = \mathbf{k}_t, \mathbf{W}_{t-s}^{(1)} = \mathbf{I}_d,$$
$$\hat{\mathbf{q}}_s^{(2)} = \mathbf{q}_s, \hat{\mathbf{k}}_t^{(2)} = \mathbf{I}_d, \mathbf{W}_{t-s}^{(2)} = \frac{1}{d}\underbrace{\left[\bar{\mathbf{k}}_{g(s-t)} \quad \cdots \quad \bar{\mathbf{k}}_{g(s-t)}\right]}_{d \text{ columns}}, \tag{35}$$
$$\hat{\mathbf{q}}_s^{(3)} = \mathbf{I}_d, \hat{\mathbf{k}}_t^{(3)} = \mathbf{k}_t, \mathbf{W}_{t-s}^{(3)} = \frac{1}{d}\underbrace{\left[\bar{\mathbf{q}}_{g(t-s)} \quad \cdots \quad \bar{\mathbf{q}}_{g(t-s)}\right]}_{d \text{ columns}}.$$

**cosFormer** (Qin et al., 2022):

$$f_{\mathrm{rel}}(\mathbf{q}_s, \mathbf{k}_t) = \mathbf{q}_s^{\mathsf{H}}\mathbf{k}_t \cos(\alpha(t-s)), \tag{36}$$

which indicates that the relative positional encoding is effectively a coefficient term in the attention matrix, as such, it can be derived via a positional matrix primitive with the coefficients.

$$m = 1,$$
$$\hat{\mathbf{q}}_s^{(1)} = \mathbf{q}_s, \hat{\mathbf{k}}_t^{(1)} = \mathbf{k}_t, \mathbf{W}_{t-s}^{(1)} = \cos(\alpha(t-s))\mathbf{I}_d. \tag{37}$$

### C.2 LINEARIZED RELATIVE POSITIONAL ENCODING

*Proof of 3.2.* For this, we only need to prove that the time complexity is linear with respect to $n$. To this end, we first give basic notations as follows,

$$\mathbf{Q} = \begin{bmatrix} \mathbf{q}_1^{\mathsf{H}} \\ \vdots \\ \mathbf{q}_n^{\mathsf{H}} \end{bmatrix} \in \mathbb{C}^{n \times d}, \mathbf{K} = \begin{bmatrix} \mathbf{k}_1^{\mathsf{H}} \\ \vdots \\ \mathbf{k}_n^{\mathsf{H}} \end{bmatrix} \in \mathbb{C}^{n \times d}, \mathbf{V} = \begin{bmatrix} \mathbf{v}_1^{\mathsf{H}} \\ \vdots \\ \mathbf{v}_n^{\mathsf{H}} \end{bmatrix} \in \mathbb{C}^{n \times d},$$
$$\tilde{\mathbf{Q}} = \begin{bmatrix} (\mathbf{M}_1\mathbf{q}_1)^{\mathsf{H}} \\ \vdots \\ (\mathbf{M}_n\mathbf{q}_n)^{\mathsf{H}} \end{bmatrix} \in \mathbb{C}^{n \times d}, \tilde{\mathbf{K}} = \begin{bmatrix} (\mathbf{M}_1\mathbf{k}_1)^{\mathsf{H}} \\ \vdots \\ (\mathbf{M}_n\mathbf{k}_n)^{\mathsf{H}} \end{bmatrix} \in \mathbb{C}^{n \times d}. \tag{38}$$

The time complexity of transforming $\mathbf{Q}, \mathbf{K}$ to $\tilde{\mathbf{Q}}, \tilde{\mathbf{K}}$ is $O(nd^2)$. The next step is to calculate the output, *i.e.*,

$$\mathbf{O} = \mathbf{Q}(\mathbf{K}^{\mathsf{H}}\mathbf{V}) \in \mathbb{C}^{n \times d},$$
$$\mathbf{O} = \mathbf{\Delta}^{-1}\tilde{\mathbf{Q}}\tilde{\mathbf{K}}^{\mathsf{H}}\mathbf{V}$$
$$= \mathbf{\Delta}^{-1}\tilde{\mathbf{Q}}[\tilde{\mathbf{K}}^{\mathsf{H}}\mathbf{V}], \tag{39}$$
$$\mathbf{\Delta} = \mathrm{diag}(\tilde{\mathbf{Q}})[\tilde{\mathbf{K}}^{\mathsf{H}}\mathbf{1}_n].$$

Clearly, Eq. 39 is a standard formulation for the linear attention with the time complexity as $O(nd^2)$. Combing it with the first step, we have the total time complexity as $O(nd^2)$, which is unchanged. □

### C.3 LINEARIZED RELATIVE POSITIONAL ENCODING

Before the proof, we first give the following theorems (Yao & Algebra, 2015):

**Theorem C.1.** *If matrix* $\mathbf{W} \in \mathbb{C}^{d \times d}$ *is a unitary matrix, there exists another **unitary** matrix* $\mathbf{P} \in \mathbb{C}^{d \times d}$*, such that*

$$
\begin{aligned}
\mathbf{W} &= \mathbf{P}^{\mathsf{H}} \mathbf{\Lambda} \mathbf{P}, \\
\mathbf{\Lambda} &= \mathrm{diag}\{\exp(i\theta_1), \dots, \exp(i\theta_d)\}, \\
i^2 &= -1.
\end{aligned}
\tag{40}
$$

**Theorem C.2.** *If matrix* $\mathbf{W} \in \mathbb{R}^{d \times d}$ *is an orthogonal matrix, there exists another **orthogonal** matrix* $\mathbf{P} \in \mathbb{R}^{d \times d}$*, such that*

$$
\begin{aligned}
\mathbf{W} &= \mathbf{P}^{\mathsf{T}} \mathbf{\Lambda} \mathbf{P}, \\
\mathbf{\Lambda} &= \mathrm{diag}\{\mathbf{\Lambda}_1, \dots, \mathbf{\Lambda}_r; 1, \dots, 1; -1, \dots, -1\}, \\
\mathbf{\Lambda}_k &= \begin{bmatrix} \cos\theta_k & -\sin\theta_k \\ \sin\theta_k & \cos\theta_k \end{bmatrix}, k = 1, \dots r.
\end{aligned}
\tag{41}
$$

## C.4 UNITARY (SOLUTION 1)

*Proof of Proposition 3.1.* According to Theorem C.1, we can assume that $\mathbf{W}_s$ has the following form ($\mathbf{P} \in \mathbb{C}^{d \times d}$ is a **unitary** matrix),

$$
\begin{aligned}
\mathbf{W}_s &= \mathbf{P}^{\mathsf{H}} \mathbf{\Lambda}^{(s)} \mathbf{P}, \\
\mathbf{\Lambda}^{(s)} &= \mathrm{diag}\{\exp(i\theta_1^{(s)}), \dots, \exp(i\theta_d^{(s)})\}.
\end{aligned}
\tag{42}
$$

Hence, Eq. 14 is equivalent to

$$
\begin{aligned}
\mathbf{W}_s^{\mathsf{H}} \mathbf{W}_t &= \mathbf{W}_{t-s}, \\
\mathbf{P}^{\mathsf{H}} \mathbf{\Lambda}^{(s)\mathsf{H}} \mathbf{P} \mathbf{P}^{\mathsf{H}} \mathbf{\Lambda}^{(t)} \mathbf{P} &= \mathbf{P}^{\mathsf{H}} \mathbf{\Lambda}^{(t-s)} \mathbf{P}, \\
\mathbf{P}^{\mathsf{H}} \mathbf{\Lambda}^{(s)\mathsf{H}} \mathbf{\Lambda}^{(t)} \mathbf{P} &= \mathbf{P}^{\mathsf{H}} \mathbf{\Lambda}^{(t-s)} \mathbf{P}, \\
\mathbf{\Lambda}^{(s)\mathsf{H}} \mathbf{\Lambda}^{(t)} &= \mathbf{\Lambda}^{(t-s)},
\end{aligned}
$$
$$
\mathrm{diag}\left\{ j(\theta_1^{(t)} - \theta_1^{(s)}), j(\theta_2^{(t)} - \theta_2^{(s)}), \cdots, j(\theta_d^{(t)} - \theta_d^{(s)}) \right\} = \mathrm{diag}\left\{ j\theta_1^{(t-s)}, j\theta_2^{(t-s)}, \cdots, j\theta_d^{(t-s)} \right\}.
\tag{43}
$$

In this case, $\forall k = 1, \dots, d$, we have

$$
\theta_k^{(t)} - \theta_k^{(s)} = \theta_k^{(t-s)} + 2k\pi, k \in \mathbb{Z}.
\tag{44}
$$

Note that $2t\pi$ does not affect the result, so we can assume $t = 0$, *i.e.*,

$$
\theta_k^{(t)} - \theta_k^{(s)} = \theta_k^{(t-s)}.
\tag{45}
$$

Taking $t = s + 1$, we get

$$
\begin{aligned}
\theta_k^{(s+1)} - \theta_k^{(s)} &= \theta_k^{(1)}, \\
\theta_k^{(s)} &= s\theta_k^{(1)} \triangleq s\alpha_k.
\end{aligned}
\tag{46}
$$

$\square$

## C.5 ORTHOGONAL (SOLUTION 2)

*Proof of Proposition 3.2.* According to Theorem C.2, we can assume that $\mathbf{W}_s$ has the following form ($\mathbf{P} \in \mathbb{R}^{d \times d}$ is an **orthogonal** matrix),

$$\mathbf{W}_s = \mathbf{P}^\mathsf{T} \mathbf{\Lambda}^{(s)} \mathbf{P},$$

$$\mathbf{\Lambda}^{(s)} = \begin{bmatrix} \mathbf{A}^{(s)} & & \\ & \mathbf{B}^{(s)} & \\ & & \mathbf{C}^{(s)} \end{bmatrix},$$

$$\mathbf{A}^{(s)} = \begin{bmatrix} \mathbf{A}_1^{(s)} & & \\ & \ddots & \\ & & \mathbf{A}_n^{(s)} \end{bmatrix} \in \mathbb{R}^{2p \times 2p}, \tag{47}$$

$$\mathbf{B}^{(s)} = \mathbf{I}_q \in \mathbb{R}^{q \times q},$$

$$\mathbf{C}^{(s)} = -\mathbf{I}_r \in \mathbb{R}^{r \times r},$$

$$\mathbf{A}_k^{(s)} = \begin{bmatrix} \cos \theta_k^{(s)} & -\sin \theta_k^{(s)} \\ \sin \theta_k^{(s)} & \cos \theta_k^{(s)} \end{bmatrix}.$$

Hence, Eq. 14 is equivalent to

$$\mathbf{W}_s^\mathsf{T} \mathbf{W}_t = \mathbf{W}_{t-s},$$

$$\mathbf{P}^\mathsf{T} \mathbf{\Lambda}^{(s)^\mathsf{T}} \mathbf{P} \mathbf{P}^\mathsf{T} \mathbf{\Lambda}^{(t)} \mathbf{P} = \mathbf{P}^\mathsf{T} \mathbf{\Lambda}^{(t-s)} \mathbf{P},$$

$$\mathbf{P}^\mathsf{T} \mathbf{\Lambda}^{(s)^\mathsf{T}} \mathbf{\Lambda}^{(t)} \mathbf{P} = \mathbf{P}^\mathsf{T} \mathbf{\Lambda}^{(t-s)} \mathbf{P},$$

$$\mathbf{\Lambda}^{(s)^\mathsf{T}} \mathbf{\Lambda}^{(t)} = \mathbf{\Lambda}^{(t-s)}, \tag{48}$$

$$\begin{bmatrix} \mathbf{A}^{(s)^\mathsf{T}} & & \\ & \mathbf{B}^{(s)^\mathsf{T}} & \\ & & \mathbf{C}^{(s)^\mathsf{T}} \end{bmatrix} \begin{bmatrix} \mathbf{A}^{(t)} & & \\ & \mathbf{B}^{(t)} & \\ & & \mathbf{C}^{(t)} \end{bmatrix} = \begin{bmatrix} \mathbf{A}^{(t-s)} & & \\ & \mathbf{B}^{(t-s)} & \\ & & \mathbf{C}^{(t-s)} \end{bmatrix},$$

where

$$\mathbf{A}^{(s)^\mathsf{T}} \mathbf{A}^{(t)} = \mathbf{A}^{(t-s)},$$

$$\mathbf{B}^{(s)^\mathsf{T}} \mathbf{B}^{(t)} = \mathbf{B}^{(t-s)}, \tag{49}$$

$$\mathbf{B}^{(s)^\mathsf{T}} \mathbf{B}^{(t)} = \mathbf{C}^{(t-s)}.$$

For $\mathbf{A}^{(s)}$, considering the $k$-th component, we get

$$\begin{aligned}
\mathbf{A}_k^{(s)^\mathsf{T}} \mathbf{A}_k^{(t)} &= \mathbf{A}_k^{(t-s)} \\
&= \begin{bmatrix} \cos \theta_k^{(s)} & \sin \theta_k^{(s)} \\ -\sin \theta_k^{(s)} & \cos \theta_k^{(s)} \end{bmatrix} \begin{bmatrix} \cos \theta_k^{(t)} & -\sin \theta_k^{(t)} \\ \sin \theta_k^{(t)} & \cos \theta_k^{(t)} \end{bmatrix} \\
&= \begin{bmatrix} \cos \theta_k^{(s)} \cos \theta_k^{(t)} + \sin \theta_k^{(s)} \cos \theta_k^{(t)} & \sin \theta_k^{(s)} \cos \theta_k^{(t)} - \cos \theta_k^{(s)} \sin \theta_k^{(t)} \\ -\sin \theta_k^{(s)} \cos \theta_k^{(t)} + \cos \theta_k^{(s)} \sin \theta_k^{(t)} & \cos \theta_k^{(s)} \cos \theta_k^{(t)} + \sin \theta_k^{(s)} \sin \theta_k^{(t)} \end{bmatrix} \\
&= \begin{bmatrix} \cos \left( \theta_k^{(t)} - \theta_k^{(s)} \right) & -\sin \left( \theta_k^{(t)} - \theta_k^{(s)} \right) \\ \sin \left( \theta_k^{(t)} - \theta_k^{(s)} \right) & \cos \left( \theta_k^{(t)} - \theta_k^{(s)} \right) \end{bmatrix} \\
&= \mathbf{A}_k^{(t-s)} \\
&= \begin{bmatrix} \cos \theta_k^{(t-s)} & -\sin \theta_k^{(t-s)} \\ \sin \theta_k^{(t-s)} & \cos \theta_k^{(t-s)} \end{bmatrix}.
\end{aligned} \tag{50}$$

Hence, $\forall k = 1, \ldots, d$, we have

$$\theta_k^{(t)} - \theta_k^{(s)} = \theta_k^{(t-s)} + 2k\pi, k \in \mathbb{Z}. \tag{51}$$

Note that $2t\pi$ does not affect the result, so we can assume $t = 0$, *i.e.*,

$$\theta_k^{(t)} - \theta_k^{(s)} = \theta_k^{(t-s)}. \tag{52}$$

Taking $t = s + 1$, we have

$$\theta_k^{(s+1)} - \theta_k^{(s)} = \theta_k^{(1)}, \tag{53}$$
$$\theta_k^{(s)} = s\theta_k^{(1)} \triangleq s\alpha_k.$$

Next, for $\mathbf{B}^{(s)}$, the conclusion is more obvious, *i.e.*,

$$\begin{aligned}
\mathbf{B}^{(s)^\mathsf{T}}\mathbf{B}^{(t)} &= \mathbf{I}_q^\mathsf{T}\mathbf{I}_q \\
&= \mathbf{I}_q \\
&= \mathbf{B}^{(t-s)}.
\end{aligned} \tag{54}$$

Finally, for $\mathbf{C}^{(s)}$, we have

$$\begin{aligned}
\mathbf{C}^{(s)^\mathsf{T}}\mathbf{C}^{(t)} &= (-\mathbf{I}_r^T)(-\mathbf{I}_r) \\
&= \mathbf{I}_r \\
&\neq \mathbf{C}^{(t-s)}.
\end{aligned} \tag{55}$$

In that case, we must have $r = 0$.

$\square$

## C.6 PERMUTATION (SOLUTION 3)

Prior to the proof, we first provide some relevant definitions and propositions.

**Definition C.3.** *Permutation $\pi$ is a **bijection** defined on the integer set:*

$$\pi : \{1, 2, \cdots, d\} \rightarrow \{1, 2, \cdots, d\}, d \in \mathbb{Z}^+. \tag{56}$$

**Definition C.4.** *For matrix*

$$\mathbf{M} = \begin{bmatrix} \mathbf{m}_1^\mathsf{T} \\ \mathbf{m}_2^\mathsf{T} \\ \vdots \\ \mathbf{m}_d^\mathsf{T} \end{bmatrix} \in \mathbb{R}^{d \times d}, \mathbf{m}_k \in \mathbb{R}^d, k = 1, \ldots, d, \tag{57}$$

$\mathbf{M}_\pi$ *is defined as*

$$\mathbf{M}_\pi = \begin{bmatrix} \mathbf{m}_{\pi(1)}^\mathsf{T} \\ \mathbf{m}_{\pi(2)}^\mathsf{T} \\ \vdots \\ \mathbf{m}_{\pi(d)}^\mathsf{T} \end{bmatrix}. \tag{58}$$

**Definition C.5.** *For identity matrix $\mathbf{I}_d \in \mathbb{R}^{d \times d}$ and permutation $\pi$, we define*

$$\mathbf{\Lambda}_k = (\mathbf{I}_d)_{\pi^k}. \tag{59}$$

For $\mathbf{\Lambda}_k$, we have the following important properties:

**Lemma C.6.** *For permutation $\pi$, matrix $\mathbf{M} \in \mathbb{R}^{d \times d}$ and $\mathbf{\Lambda}_k \in \mathbb{R}^{d \times d}$ defined in C.5, we have*

$$\mathbf{M}_\pi = \mathbf{\Lambda}_1 \mathbf{M}. \tag{60}$$

*Proof.* We first organize $\mathbf{I}_d \in \mathbb{R}^{d \times d}$ in the following form, where $\mathbf{e}_k \in \mathbb{R}^d, k = 1, \ldots, d$ represents the one-hot vector with the $k$-th element as one, *i.e.*,

$$\mathbf{I}_d = \begin{bmatrix} \mathbf{e}_1^\mathsf{T} \\ \mathbf{e}_2^\mathsf{T} \\ \vdots \\ \mathbf{e}_d^\mathsf{T} \end{bmatrix}. \tag{61}$$

Notice that

$$\mathbf{e}_k^\mathsf{T}\mathbf{M} = \mathbf{m}_k^\mathsf{T}, \tag{62}$$

so we get

$$
\begin{aligned}
\mathbf{\Lambda}_1\mathbf{M} &= \begin{bmatrix} \mathbf{e}_{\pi(1)}^\mathsf{T} \\ \mathbf{e}_{\pi(2)}^\mathsf{T} \\ \vdots \\ \mathbf{e}_{\pi(d)}^\mathsf{T} \end{bmatrix} \mathbf{M} \\
&= \begin{bmatrix} \mathbf{e}_{\pi(1)}^\mathsf{T}\mathbf{M} \\ \mathbf{e}_{\pi(2)}^\mathsf{T}\mathbf{M} \\ \vdots \\ \mathbf{e}_{\pi(d)}^\mathsf{T}\mathbf{M} \end{bmatrix} \\
&= \begin{bmatrix} \mathbf{m}_{\pi(1)}^\mathsf{T} \\ \mathbf{m}_{\pi(2)}^\mathsf{T} \\ \vdots \\ \mathbf{m}_{\pi(d)}^\mathsf{T} \end{bmatrix} \\
&= \mathbf{M}_\pi.
\end{aligned} \tag{63}
$$

$\square$

**Theorem C.7.** *For $\mathbf{\Lambda}_k$ defined in C.5, we have:*

$$\mathbf{\Lambda}_k = \mathbf{\Lambda}_1^k. \tag{64}$$

*Proof.* We use induction for the proof.

For $k = 1$, the conclusion is obvious. Now assuming that the conclusion holds for $k = s - 1$, when $k = s$, we have

$$
\begin{aligned}
\mathbf{\Lambda}_s &= (\mathbf{I}_d)_{\pi^s} \\
&= ((\mathbf{I}_d)_{\pi^{s-1}})_\pi \\
&= (\mathbf{\Lambda}_{s-1})_\pi \\
&= (\mathbf{\Lambda}_1^{s-1})_\pi.
\end{aligned} \tag{65}
$$

The next step is to prove

$$(\mathbf{\Lambda}_1^{s-1})_\pi = \mathbf{\Lambda}_1^s = \mathbf{\Lambda}_1\mathbf{\Lambda}_1^{s-1}. \tag{66}$$

The above conclusion follows from C.6.

$\square$

**Theorem C.8.** $\mathbf{\Lambda}_k \in \mathbb{R}^{d\times d}$ *defined in C.5 are orthogonal matrices, i.e.,*

$$\mathbf{\Lambda}_k\mathbf{\Lambda}_k^T = \mathbf{\Lambda}_k^\mathsf{T}\mathbf{\Lambda}_k = \mathbf{I}_d. \tag{67}$$

*Proof.* We first prove that the conclusion holds for $k = 1$:

$$
\begin{aligned}
\mathbf{\Lambda}_1\mathbf{\Lambda}_1^\mathsf{T} &= \begin{bmatrix} \mathbf{e}_{\pi(1)}^\mathsf{T} \\ \mathbf{e}_{\pi(2)}^\mathsf{T} \\ \vdots \\ \mathbf{e}_{\pi(d)}^\mathsf{T} \end{bmatrix} \begin{bmatrix} \mathbf{e}_{\pi(1)} & \mathbf{e}_{\pi(2)} & \cdots & \mathbf{e}_{\pi(d)} \end{bmatrix}, \\
\left[\mathbf{\Lambda}_1\mathbf{\Lambda}_1^\mathsf{T}\right]_{st} &= \mathbf{e}_{\pi(s)}^\mathsf{T}\mathbf{e}_{\pi(t)} \\
&= \delta_{st}, \\
\mathbf{\Lambda}_1\mathbf{\Lambda}_1^\mathsf{T} &= \mathbf{I}_d.
\end{aligned} \tag{68}
$$

Since $\boldsymbol{\Lambda}_1$ is a square matrix, we also have

$$\boldsymbol{\Lambda}_1^{\mathsf{T}}\boldsymbol{\Lambda}_1 = \mathbf{I}_d. \tag{69}$$

In general cases, we only use C.7, *i.e.*,

$$\begin{aligned}
\boldsymbol{\Lambda}_k\boldsymbol{\Lambda}_k^{\mathsf{T}} &= \boldsymbol{\Lambda}_1^k(\boldsymbol{\Lambda}_1^k)^{\mathsf{T}} \\
&= \boldsymbol{\Lambda}_1^k(\boldsymbol{\Lambda}_1^{\mathsf{T}})^k \\
&= \boldsymbol{\Lambda}_1^{k-1}\boldsymbol{\Lambda}_1\boldsymbol{\Lambda}_1^{\mathsf{T}}(\boldsymbol{\Lambda}_1^{\mathsf{T}})^{k-1} \\
&= \boldsymbol{\Lambda}_1^{k-1}(\boldsymbol{\Lambda}_1^{\mathsf{T}})^{k-1} \\
&= \dots \\
&= \mathbf{I}_d.
\end{aligned} \tag{70}$$

With the same proof, we get

$$\boldsymbol{\Lambda}_k^{\mathsf{T}}\boldsymbol{\Lambda}_k = \mathbf{I}_d. \tag{71}$$

$\square$

Based on the above conclusions, we can prove Proposition 3.3 below.

*Proof of Proposition 3.3.* According to Theorem C.8 and the production of the **orthogonal** matrix is an **orthogonal** matrix, we can assume that $\mathbf{W}_k$ has the following form ($\mathbf{P} \in \mathbb{R}^{d \times d}$ is an **orthogonal** matrix), *i.e.*,

$$\mathbf{W}_k = \mathbf{P}^{\mathsf{T}}\boldsymbol{\Lambda}^{(k)}\mathbf{P}. \tag{72}$$

The next step is to verify that it satisfies Eq. 14, which follows Theorem C.7 and C.8:

$$\begin{aligned}
\mathbf{W}_s^{\mathsf{T}}\mathbf{W}_t &= \mathbf{P}^{\mathsf{T}}\boldsymbol{\Lambda}^{(s)^{\mathsf{T}}}\mathbf{P}\mathbf{P}^{\mathsf{T}}\boldsymbol{\Lambda}^{(t)}\mathbf{P} \\
&= \mathbf{P}^{\mathsf{T}}\boldsymbol{\Lambda}^{(s)^{\mathsf{T}}}\boldsymbol{\Lambda}^{(t)}\mathbf{P} \\
&= \mathbf{P}^{\mathsf{T}}\boldsymbol{\Lambda}^{(s)^{\mathsf{T}}}(\boldsymbol{\Lambda}^{(1)})^t\mathbf{P} \\
&= \mathbf{P}^{\mathsf{T}}\boldsymbol{\Lambda}^{(s)^{\mathsf{T}}}(\boldsymbol{\Lambda}^{(1)})^s(\boldsymbol{\Lambda}^{(1)})^{t-s}\mathbf{P} \\
&= \mathbf{P}^{\mathsf{T}}\boldsymbol{\Lambda}^{(s)^{\mathsf{T}}}\boldsymbol{\Lambda}^{(s)}(\boldsymbol{\Lambda}^{(1)})^{t-s}\mathbf{P} \\
&= \mathbf{P}^{\mathsf{T}}\boldsymbol{\Lambda}^{(t-s)}\mathbf{P} \\
&= \mathbf{W}_{t-s}.
\end{aligned} \tag{73}$$

$\square$

## C.7 IMPLEMENTATION

LRPE($\mathbf{W}_s = \mathbf{P}^{\mathsf{H}}\boldsymbol{\Lambda}^{(s)}\mathbf{P}$) contains two components, *i.e.*, the fixed unitary matrix $\mathbf{P}$ and the unitary matrix family $\boldsymbol{\Lambda}^{(s)}$ mentioned in proposition 3.1, 3.2, and 3.3. We first introduce the choice of matrices $\mathbf{P}/\boldsymbol{\Lambda}^{(s)}$, and then illustrate some implementation tricks.

**Choice of matrices**

For matrix $\mathbf{P}$, we employ three types as follows,

- Householder matrix: denoted as a vector $\mathbf{v} \in \mathbb{R}^d$, *i.e.*,

$$\mathbf{W} = \mathbf{I}_d - 2\mathbf{v}\mathbf{v}^{\mathsf{T}}/(\mathbf{v}^{\mathsf{T}}\mathbf{v}). \tag{74}$$

  In our implementation, we sample $\mathbf{v}$ from standard normal distribution, and make it **deterministic** or **learnable**.

- Permutation matrix: formulated as per the following permutation (inspired by Flash (Hua et al., 2022)), *i.e.*,

$$\pi(2k) = k, \pi(2k+1) = \lfloor d/2 \rfloor + 1, 1 \le 2k, 2k+1 \le d. \tag{75}$$

- FFT matrix: a matrix form of FFT (Fast Fourier Transform).

For **matrix family** $\mathbf{\Lambda}^{(s)}$, we use the following settings:

- For unitary (Solution 1) (3.1), we use the same method in (Su et al., 2021) with initialized $\alpha_t = 10000^{-2t/d}$, and make it deterministic. Since this method involves complex numbers, we **only** use the FFT matrix for the choice of $\mathbf{P}$.

- For orthogonal (Solution 2) (3.2), we test with two versions. In the first version, we set the dimension of the identity submatrix $q = \lfloor d/2 \rfloor$, initialized $\alpha_t = 10000^{-2t/d}$ as in (Su et al., 2021) and make it **deterministic**. In the second version, we choose the dimension of identity submatrix $q = 0$ with initialized $\alpha_t = 10000^{-2t/d}$ as in (Su et al., 2021), and make it **learnable**.

  - Another notable version to choose the dimension of the identity submatrix $q = 0$ with initialized $\alpha_t = 10000^{-2t/d}$ as in (Su et al., 2021), and make it **deterministic**. When using this version along with the identity matrix, we can get **RoPE** (Su et al., 2021).

- For permutation (Solution 3) (3.3), we randomly choose the permutation and make it **deterministic**.

  - Notice that when combing this method with identity matrix, we can get a version of **PermutateFormer** (Chen, 2021).

**Implementation tricks**

According to the following facts, we can simplify the computation, *i.e.*,

$$
\begin{aligned}
\mathbf{q}_s^{\mathsf{H}} \mathbf{W}_s^{\mathsf{H}} \mathbf{W}_t \mathbf{k}_t &= \mathbf{q}_s^{\mathsf{H}} \mathbf{P}^{\mathsf{H}} (\mathbf{\Lambda}^{(s)})^{\mathsf{H}} \mathbf{P} \mathbf{P}^{\mathsf{H}} \mathbf{\Lambda}^{(t)} \mathbf{P} \mathbf{k}_t \\
&= \mathbf{q}_s^{\mathsf{H}} \mathbf{P}^{\mathsf{H}} (\mathbf{\Lambda}^{(s)})^{\mathsf{H}} \mathbf{\Lambda}^{(t)} \mathbf{P} \mathbf{k}_t \\
&= (\mathbf{\Lambda}^{(s)} \mathbf{P} \mathbf{q}_s)^{\mathsf{H}} (\mathbf{\Lambda}^{(t)} \mathbf{P} \mathbf{k}_t).
\end{aligned}
\tag{76}
$$

Hence, in practice, we can use $\mathbf{W}_s = \mathbf{P}^{\mathsf{H}} \mathbf{\Lambda}^{(s)}$ instead of $\mathbf{W}_s = \mathbf{P}^{\mathsf{H}} \mathbf{\Lambda}^{(s)} \mathbf{P}$ to reduce the computational costs.

## C.8 PSEUDOCODE

In this section, we provide pseudocodes for LRPE in Python:

```python
import torch
import torch.nn as nn
import numpy as np

class Lrpe(nn.Module):
    def __init__(self, core_matrix, p_matrix, max_positions=512,
        embedding_dim=768,
                theta_type="a", theta_learned=False,
                    householder_learned=False):
        super().__init__()
        self.core_matrix = core_matrix
        self.p_matrix = p_matrix
        self.theta_type = theta_type
        self.theta_learned = theta_learned
        self.householder_learned = householder_learned

        # Lambda matrix
        if self.core_matrix == 1:
            if self.theta_learned:
                print("Learn theta!")
                self.theta = nn.Parameter(10000 ** (-2 / embedding_dim *
                    torch.arange(embedding_dim // 2)).reshape(1, 1, -1))
            else:
                print(f"Theta_type {self.theta_type}")
```

```python
        elif self.core_matrix == 2:
            print("Mixed")
        elif self.core_matrix == 3:
            print("Permutation")
            permutation = self.get_permutation(max_positions, embedding_dim)
            self.register_buffer("permutation", permutation)
        elif self.core_matrix == 4:
            print("Complex exp")
            if self.theta_learned:
                print("Learn theta!")
                self.theta = nn.Parameter(10000 ** (-2 / embedding_dim *
                    torch.arange(embedding_dim)).reshape(1, 1, -1))
            else:
                print(f"Theta_type {self.theta_type}")

        # P matrix
        if self.p_matrix == 1:
            print("Identity")
        elif self.p_matrix == 2:
            print("Householder")
            if self.householder_learned:
                print("learn householder!")
                self.v = nn.Parameter(torch.randn(1, embedding_dim, 1))
            else:
                v = torch.randn(1, embedding_dim, 1)
                v = v / torch.norm(v)
                print(f"Householder norm is {torch.norm(v)}")
                self.v = nn.Parameter(v, requires_grad=False)
        elif self.p_matrix == 3:
            print("Fourier")
        elif self.p_matrix == 4:
            print("Odd_even")

        self.p = self.get_p()
        self.core_transform = self.get_core_transform()

    def forward(self, x):
        '''
        input shape: (b, l, e), b stands for batch size, l stands for
            sequence length, e stands for embedding dimension.
        '''
        x = self.p(x)
        x = self.core_transform(x)
        return x

    def get_p(self):
        if self.p_matrix == 1:
            def f(x):
                return x
            return f
        elif self.p_matrix == 2:
            return self.householder
        elif self.p_matrix == 3:
            def f(x):
                return torch.fft.fft(x, norm="ortho")
            return f
        elif self.p_matrix == 4:
            return self.odd_even_permutation

    def get_core_transform(self):
        if self.core_matrix == 1:
            return self.reflect
        elif self.core_matrix == 2:
            return self.mix_reflect
        elif self.core_matrix == 3:
```

```python
            return self.do_permutation
        elif self.core_matrix == 4:
            return self.complex_exp

    def get_permutation(self, max_positions, embedding_dim):
        permutation = torch.randperm(embedding_dim).reshape(1, -1)
        expanded = [torch.arange(embedding_dim).unsqueeze(0)]
        for _ in range(max_positions - 1):
            previous = expanded[-1]
            current = previous.gather(-1, permutation)
            expanded.append(current)
        expanded = torch.stack(expanded, dim=1)
        return expanded

    def odd_even_permutation(self, x):
        # 2k->k, 2k+1->d+k
        e = x.shape[-1]
        d = e - e // 2
        permutation = torch.arange(e)
        index = torch.arange(e)
        permutation[::2] = index[::2] // 2
        permutation[1::2] = (index[1::2] - 1) // 2 + d
        permutation = permutation.to(x.device)
        x = x.gather(-1, permutation.expand_as(x))

        return x

    def do_permutation(self, x):
        b, l, e = x.shape
        x = x.gather(-1, self.permutation[:, :l, :].expand_as(x))

        return x

    def reflect(self, x):
        b, l, d = x.shape
        e = d - 1 if d % 2 == 1 else d
        return self.transform(x, e)

    def mix_reflect(self, x):
        b, l, d = x.shape
        assert d >= 3
        # split
        e = d // 2
        # to even
        if e % 2:
            e += 1
        return self.transform(x, e)

    def transform(self, x, e):
        assert e % 2 == 0
        b, l, d = x.shape
        # do identity transformation
        x1 = x[:, :, e:]
        # do reflection
        x = x[:, :, :e]
        if self.theta_learned:
            theta = self.theta
        else:
            if self.theta_type == "a":
                theta = 10000 ** (-2 / e * torch.arange(e // 2))
            elif self.theta_type == "b":
                theta = np.pi / 2 / l / (e // 2) * torch.arange(1, e // 2 +
                    1)
            elif self.theta_type == "c":
                theta = np.pi / 2 / l / torch.arange(1, e // 2 + 1)
```

```python
        theta = theta.reshape(1, 1, -1).to(x)
    theta = torch.stack([theta, theta], dim=-1).reshape(1, 1, e)
    theta = theta * torch.arange(l).reshape(1, -1, 1).to(x)
    # (-q1, -q3), (q0, q2) -> (-q1, q0, -q3, q2)
    x_half = torch.stack([-x[..., 1::2], x[..., ::2]],
        dim=-1).reshape_as(x)
    x_transform = x * torch.cos(theta) + x_half * torch.sin(theta)
    # merge
    if e != d:
        x_transform = torch.cat([x_transform, x1], dim=-1)

    return x_transform

def complex_exp(self, x):
    b, l, e = x.shape
    if self.theta_learned:
        theta = self.theta
    else:
        if self.theta_type == "a":
            theta = 10000 ** (-2 / e * torch.arange(e))
        theta = theta.reshape(1, 1, -1).to(x.device)
    matrix = theta * torch.arange(l).reshape(1, -1, 1).to(x.device)

    sin_cos =
        torch.complex(torch.cos(matrix),torch.sin(matrix)).to(x.device)
    x = self.element_wise_complex(x, sin_cos)
    return x

def element_wise_complex(self, t1, t2):
    return torch.complex(t1.real * t2.real - t1.imag * t2.imag,
        t1.real * t2.imag + t1.imag * t2.real)

def householder(self, x, eps=1e-6):
    if self.householder_learned:
        v = self.v / (torch.norm(self.v) + eps)
    else:
        v = self.v
    # (b, n, e), (1, e, 1) -> (1, n, 1)
    y = torch.matmul(x, v)
    # (1, n, 1), (1, 1, e) -> (1, n, e)
    y = torch.matmul(y, v.transpose(1, 2))

    return x - 2 * y
```

# D  EXPERIMENT

## D.1  CONFIGURATION

We provide detailed data, model and training configurations in Table 7. For published datasets, WikiText-103 is obtained from https://www.salesforce.com/products/einstein/ai-research/the-wikitext-dependency-language-modeling-dataset/, with Creative Commons Attribution-ShareAlike License. The GLUE dataset is obtained from https://gluebenchmark.com/. The WMT'14 EN-DE dataset is downloaded from https://www.statmt.org/wmt14/:

## D.2  RESULTS OF BIDIRECTIONAL LANGUAGE MODEL

We report the pretrained results of the bidirectional language model in Table 8. Our LRPE achieves competitive performance in both linear attention and vanilla attention. Notably, the UPPE variant $UPRE\_ol\_h$ has the best quantitative results in all evaluation metrics.

Table 7: Detailed configurations used in our experiments. "Total batch size" means batch_per_gpu × update_freq × num_gpus. "Attention dropout" is only used for vanilla attention. "ALM": autoregressive Language Model. "BLM": bidirectional Language Model. "MT": Machine Translation.

| | AML | BLM | MT |
|---|---|---|---|
| Data | WikiText-103 | WikiText-103 | WMT14 EN-DE |
| Tokenizer method | BPE | BPE | BPE |
| Src Vocab size | 267744 | 50265 | 40480 |
| Tgt Vocab size | - | - | 42720 |
| Encoder layers | 0 | 12 | 6 |
| Decoder layers | 6 | 0 | 6 |
| Hidden dimensions | 512 | 768 | 512 |
| Number of heads | 8 | 12 | 8 |
| FFN dimensions | 2048 | 3072 | 2048 |
| FFN activation function | Relu | Gelu | Relu |
| Seqence length | 512 | 512 | / |
| Total batch size | 128 | 512 | / |
| Max token per batch | / | / | 4400 |
| Number of updates | 50k | 50k | 100k |
| Warmup steps | 4k | 3k | 4k |
| Peak learning rate | 5e-4 | 5e-4 | 7e-4 |
| Learning rate scheduler | Inverse sqrt | Polynomial decay | Inverse sqrt |
| Optimizer | Adam | Adam | Adam |
| Adam $\epsilon$ | 1e-8 | 1e-6 | 1e-8 |
| Adam $(\beta_1, \beta_2)$ | (0.9, 0.98) | (0.9, 0.98) | (0.9, 0.98) |
| Weight decay | 0.01 | 0.01 | 0 |
| Gradient clipping | 0.0(1.0 for Type8) | 0 | 0 |
| Hidden dropouut | 0.1 | 0.1 | 0.1 |
| Attention dropout | 0 | 0.1 | 0 |

Table 8: Quantitative results of the Roberta model pretrained on the WikiText-103 dataset. The best result is highlighted with **bold** and the second with underlined. ↓ means *smaller is better*.

| Method | | Linear attention | | Vanilla attention | |
|---|---|---|---|---|---|
| | | Loss (val)↓ | PPL (val)↓ | Loss (val)↓ | PPL (val)↓ |
| **Competitors** | Base | 2.32 | 4.98 | 1.92 | 3.77 |
| | RoPE | 2.21 | 4.64 | 1.85 | 3.60 |
| | SPE | 2.74 | 6.68 | 2.04 | 4.11 |
| | PER | 2.54 | 5.81 | - | - |
| | T5 | - | - | 1.90 | 3.72 |
| **Householder** | Type1 | 2.34 | 5.06 | 1.83 | 3.55 |
| | Type2 | **2.08** | **4.22** | **1.82** | **3.53** |
| | Type3 | 2.09 | 4.26 | 1.83 | 3.55 |
| | Type4 | 2.31 | 4.95 | 2.02 | 4.06 |
| **Permutation** | Type5 | 2.29 | 4.90 | 1.85 | 3.61 |
| | Type6 | 2.11 | 4.30 | 1.82 | 3.54 |
| | Type7 | 2.27 | 4.81 | 2.07 | 4.18 |
| **FFT** | Type8 | - | - | 1.88 | 3.68 |

## D.3 EFFICIENCY

We compare the training speed of LRPE with other methods in Table D.2, which indicates that our method maintains good efficiency without incurring too much computational burden.

Table 9: Training speed of different methods on the bidirectional language model. The value standards for the speed relative to the base method. ↑ means *larger is faster*.

| Method | Linear attention | Vanilla attention |
|---|---|---|
| | Relative speed↑ | Relative speed↑ |
| Base (Vaswani et al., 2017) | 1.00 | 1.00 |
| RoPE (Su et al., 2021) | 0.95 | 0.97 |
| SPE (Liutkus et al., 2021) | 0.42 | 0.41 |
| PER (Chen, 2021) | 0.88 | - |
| T5 (Raffel et al., 2019) | - | 0.70 |
| LRPE | 0.91 | 0.95 |

