# OpenReview forum: "Relative Positional Encoding Family via Unitary Transformation"
_ICLR.cc/2023/Conference — Submitted to ICLR 2023_

### Official Review · Reviewer_e2DD · 2022-10-22

**Confidence:** 5
**Correctness:** 2
**Technical Novelty And Significance:** 2
**Empirical Novelty And Significance:** 2
**Recommendation:** 5

**Clarity, Quality, Novelty And Reproducibility:**

The paper is clear though the significance of the performance and the novelty remain unclear.

**Strength And Weaknesses:**

Strength:
- The unified form of relative positional encoding is meaningful in the sense that this formulation makes things clear and helps to design new relative positional encoding schemes.
- The examples in this paper provide a diverse view of how to design relative positional encoding and decomposable relative positional encodings.
Weakness:
- Though the paper is motivated by designing relative positional encoding for linear attention, the empirical performance on linear attention is rather weak. Meanwhile, the paper claims that the problem of adding relative positional encoding to linear attention is under-studied, which is not true even though the paper has cited some of the related work. Concretely, there exist several works studying this problem, including [1, 2, 3] where [2] has significantly better performance than the proposed method (e.g., GLUE). Also to study linear attention, it is important to report the running time, it would be great to add discussion on this (though I understand the additional overhead is small for the method proposed in this paper)
- Some of the numbers of the baseline methods is too low. For, example, it is easy to reach 80+ on MNLI, 30- on Wikitext-103.

[1] Relative Positional Encoding for Transformers with Linear Complexity
[2] Stable, Fast and Accurate: Kernelized Attention with Relative Positional Encoding
[3] PermuteFormer: Efficient Relative Position Encoding for Long Sequences

**Summary Of The Paper:**

This paper proposed to unify relative positional encoding methods by formulating them as a quadratic form and introduced a class of decomposable relative positional encoding that is equipable by linear attention.

**Summary Of The Review:**

In summary, I think the unified view of relative positional encoding is interesting, but the results in the paper are not strong enough. Also, the baselines and reviews in the paper are not comprehensive enough.

---

> ### Author Response · Authors · 2022-11-18
> **Response to  e2DD(Part 1)**
>
> **Q1: Research about combing linear attention with relative position encoding.**
>
> We compared the mentioned methods [1], [3] in Tables 2, 3, 4, denoted by SPE and PER, respectively. In most cases, our method achieves better performance than these two methods.
>
> For the mentioned method [2], denoted by Krpe, we conduct an additional comparison under our configuration on Lm, Nmt tasks. The results are shown below. Note that we do not directly compare the results reported in the paper [2], as the configuration and training corpus are totally different. For example,
>
> - The pre-training corpus used by Krpe a 160G corpus (see appendix A of [2]), while we used the wikitext-103 dataset (see appendix D of our paper) which is only 183.09 Mb (see [hyperlink](https://huggingface.co/datasets/wikitext) for detailed information).
> - The number of pre-training epochs and batch size used by Krpe are 1000k and 2048, respectively. We update 50k times with a batch size of 512 (mentioned in Table 7);
>
> Lm:
>
> | method | ppl(valid) | ppl(test) |
> | ------ | ---------- | --------- |
> | linear | 33.94      | 33.74     |
> | Type2  | 32.83      | 32.8      |
> | krpe   | 31.77      | 31.14     |
>
> Nmt(the experiments are run 5 five times, and report the average result and standard derivation):
>
> | method | BLEU(val) | BLEU(test) |
> | ------ | --------- | ---------- |
> | linear | $29.57\pm0.06$         | $26.34\pm 1.05$          |
> | Type4  | $29.78\pm 0.05$         | $27.47\pm0.45$          |
> | krpe   | $29.87\pm 0.04$         | $26.84\pm 0.21$         |
>
> We find that Krpe has more advantages on Lm, while our method works better on Nmt.
>
> For further comparison, we analyze the theoretical and practical time complexity of Krpe.
>
> Let’s first analyze the theoretical time complexity of Krpe. The main calculation of Krpe is reflected in the following formula:
>
> $$
> \tilde{D}_1= AB  \newline,\tilde{D}_2=AC \newline
> $$
>
> and,
>
> $$
> \begin{aligned}
> A&=
> \begin{bmatrix}
> c_0 & c_1 & c_2 & \cdots & c_{n-1} \newline
> c_{-1} & c_0 & c_1 & \cdots & c_{n-2} \newline
> \vdots & \vdots & \vdots & \ddots & \vdots\newline
> c_{-(n-1)} & c_{-(n-2)} & c_{-(n-3)} & \cdots & c_0
> \end{bmatrix} \in \mathbb R^{n\times n} \newline
> B&=
> \begin{bmatrix}
> \mathrm{vec}\left(\phi\left(x_1 W^K\right)^{\top}\left(x_1 W^V\right)\right) \newline
> \mathrm{vec}\left(\phi \left(x_2 W^K\right)^{\top}\left(x_2 W^V\right)\right) \newline
> \vdots \newline
> \mathrm{vec}
> \left(\phi\left(x_n W^K\right)^{\top}\left(x_n W^V\right)\right)
> \end{bmatrix} \in \mathbb R^{n\times d^2} \newline
> D&=
> \begin{bmatrix}\mathrm{vec}\left(\phi\left(x_1 W^K\right)^{\top}\right) \newline
>     \mathrm{vec}\left(\phi\left(x_2 W^K\right)^{\top}\right) \newline
>     \vdots \newline
>     \mathrm{vec}\left(\phi\left(x_n W^K\right)^{\top}\right)
> \end{bmatrix} \in \mathbb R^{n\times d}
> \end{aligned}
> $$
>
> For Toeplitz matrix-vector production $Tx$, where $T\in \mathbb R^{n\times n}, x\in \mathbb R^{n}$, the time complexity is $O(n\log n)$. So the above operation has time complexity $O(nd^2 \log n)$.
>
> For our method, the key operation is:
>
> $$
> \mathbf q_s \to \mathbf q_s \mathbf P \Lambda_s, \mathbf k_t \to \mathbf k_t \mathbf P \Lambda_t,
> \mathbf q_s\in \mathbb R^{d}, \mathbf P \in \mathbb R^{d\times d}, \Lambda_s \in \mathbb R^{d\times d}
> $$
>
> with an time complexity of $O(nd^2)$.
>
> Next, let's analyze its practice efficiency. We test one of our methods on the bidirectional language model (Roberta). The value standards for the speed relative to the base method, which is the larger, the better:
>
> | Method | Linear |
> | ------ | ------ |
> | Base   | 1      |
> | LRPE   | 0.91   |
> | Krpe   | 0.5    |
>
> As shown above, our method performs similarly to Krpe while being significantly faster, both theoretically and practically.
>
> **Q2. Time complexity analysis**
>
> We analyzed the practical complexity of LRPE in Section 4.3 and Table 9 in Appendix D, and compared it with other methods. Here, we show our results again:
>
> | Method | Linear | Vanilla |
> | ------ | ------ | ------- |
> | Base   | 1      | 1       |
> | RoPE   | 0.95   | 0.97    |
> | SPE    | 0.42   | 0.41    |
> | PER    | 0.88   | -       |
> | T5     | -      | 0.7     |
> | LRPE   | 0.91   | 0.95    |
> | Krpe   | 0.5    | -       |
>
> In general, LRPE does not incur a significant computational burden to the transformer and can fulfill the practical needs by maintaining comparable efficiency.

---

> > ### Comment · Reviewer_e2DD · 2022-11-22
> > **Thanks for the response**
> >
> > Thanks for conducting more experiments! The new results make the paper stronger, yet I recommend using a more standard setting to compare with other methods, e.g., use at least the ~16GB BERT pre-training corpus. Otherwise, it is hard to judge the number whether the number is due to poor implementation or just because of the experimental setting. Overall, I think the paper makes good contributions but the empirical results are not strong enough. I will raise my score to 5 accordingly.

---

> ### Author Response · Authors · 2022-11-18
> **Response to e2DD(Part 2)**
>
> **Q3. About baseline**
>
> Our experiments are carried out under the condition of strictly controlling the variables, that is, controlling the model size, batch size, number of training rounds, training hyperparameters, etc., in order to fairly compare the performance of each relative positional encoding. For our experimental results, we should look at the increment relative to the baseline, not the value.
>
> For the result reported in [2], denoted by Krpe, we think it is unfair to compare our results with it for the following reasons:
>
> - For MNLI (Bert) result, please refer to Q1.
> - For Lm, [2] training the model with batch size 64 and update times 150k, while our model training the model with batch size 128 and update times 100k. For a fair comparison, we reran Krpe under the same configuration as our method, please refer to Q1 for detailed information.
>
> Citations:
>
> - [1] Relative Positional Encoding for Transformers with Linear Complexity
> - [2] Stable, Fast and Accurate: Kernelized Attention with Relative Positional Encoding
> - [3] PermuteFormer: Efficient Relative Position Encoding for Long Sequences

---

### Official Review · Reviewer_YduU · 2022-10-24

**Confidence:** 4
**Correctness:** 3
**Technical Novelty And Significance:** 2
**Empirical Novelty And Significance:** 3
**Recommendation:** 6

**Clarity, Quality, Novelty And Reproducibility:**

The paper is clear enough, and the quality is good. My main concern is novelty, as I wrote. I think it is fair though, because the paper *does* bring yet another highlight on the topic. Still, my point is that maybe this is not sufficient for such a selective conference as ICLR, justifying my score.

**Strength And Weaknesses:**

The paper provides a good overview (although a bit biased, since I believe that several previous works already highlighted the SVD thing). The proposed method makes sense, and I didn't check the implementation, but it is claimed that it will be made open.
Then, experiments are fine, although I believe that:
*  it would have been interesting to try everything with "no positional encoding". I indeed found out that it is quite common that PE is useless altogether for some tasks.
* It is likely in my opinion that the other methods would probably have given better results if tuned. But it's ok I guess that using the default parameters for their implementations is fine.

The main weakness I can see is novelty. I am really not convinced that factorizing the relative weights through SVD and then designing some particular cases is so new. It is claimed that this method subsumes most state of the art (indeed it's the case, necessarily), but it would have been good to highlight in what sense. For instance, "the method from [xxx] is equivalent to picking P as ... and \Lambda as ...". Another weakness is that computing time / memory usage is not reported very well.

**Summary Of The Paper:**

The paper focuses on the problem of designing relative positional encoding schemes that are compatible with linear transformer architectures. This problem attracted some research already, and the main idea of the paper is to express the problem in a general way, by putting some emphasis on the fact that linear complexity requires considering the relative weighting through its SVD. Doing so, they derive some variants, based on how they design the matrices involved. Performance is on par with other existing schemes, and a bit better sometimes.

**Summary Of The Review:**

another paper on relative positional encoding with linear complexity, whose highlight is on the factorization of the relative attention weight through SVD. The paper provides good background, although could be a bit more objective since I believe that the proposed method it is almost equivalent to several other previously proposed ones,.
All in all, a good paper on a topic I'm interested in, with borderline+ contribution, but still a nice read.

---

> ### Author Response · Authors · 2022-11-18
> **Response to Reviewer YduU**
>
> **Q1. Novelty and the connection between SVD and LRPE**
>
> In this paper, we provide a unified solution for encoding relative position information to linear transformers. Researchers can form infinite types of relative positional encoding methods based on the unified solution. On the other hand, previous papers often only provide one special positional encoding solution.  In light of the relationship between SVD and LRPE, even though the final form, i.e. Eq. 14 and SVD are relatively similar, their intrinsic representations are completely different. One obvious difference is that the SVD decomposition $\mathbf{M}=\mathbf{U} \boldsymbol{\Sigma} \mathbf{V}^{\mathbf H}$, requires $\mathbf \Sigma$ to be a diagonal matrix, while our method, except solution 1, does not satisfy this property.
>
> **Q2. No positional encoding experiments**
>
> As suggested, we conduct experiments without position encoding on all tasks, and the results are as follows.
>
> Lm:
>
> | Method            | PPL(val) | PPL(test) |
> | ----------------- | -------- | --------- |
> | Linear Attention  |          |           |
> | base              | 33.94    | 33.74     |
> | no pe             | 35.81    | 35.46     |
> | Vanilla Attention |          |           |
> | base              | 30.82    | 29.78     |
> | no pe             | 31.77    | 31.84     |
>
> Nmt(the experiments are run 5 five times, and report the average result and standard derivation):
>
> | Method            | BLEU(valid) | BLEU(test) |
> | ----------------- | ----------- | ---------- |
> | Linear Attention  |             |            |
> | base              | $29.57\pm0.06$          | $26.34\pm 1.05$          |
> | no pe             | $17.61\pm 0.04$           | $10.91\pm0.19$        |
> | Vanilla Attention |             |            |
> | base              | $29.92\pm 0.3$           | $27.50\pm 0.18$         |
> | no pe             | $18.26\pm0.002$           | $12.21\pm0.02$          |
>
> Roberta pretraining:
>
> | Method            | Loss(valid) | PPL(valid) |
> | ----------------- | ----------- | ---------- |
> | Linear Attention  |             |            |
> | base              | 2.317       | 4.98       |
> | no pe             | 8.42        | 342.37     |
> | Vanilla Attention |             |            |
> | base              | 1.915       | 3.77       |
> | no pe             | 8.402       | 338.16     |
>
> According to the above experimental results, it can be seen that positional encoding significantly improves the performance in the autoregressive language model, bidirectional language model, and machine translation task.
>
> **Q3. About Configuration**
>
> Here we use the same training parameters for each method, including the number of updates, batch size, learning rate, optimizer, etc. The main purpose to do this is to verify the effectiveness of these methods under the same conditions.
>
> **Q4. Relation with other methods**
>
> Our RPE family indeed includes some existing RPE methods, such as:
>
> - If we take $\mathbf P$ as the identity matrix and $\Lambda^{(s)}$ as the blocked rotation matrix (solution 1), we can get RoPE;
> - If we first transform $\mathbf q_s$ to $\lambda^s \mathbf q_s$, and transformer $\mathbf k_t$ to $\lambda^{-t}\mathbf k_t$ and take $\mathbf P$ as identity matrix and $\Lambda^{(s)}$ as permutation matrix (solution 3), we can get PermutateFormer.
>
> **Q5. Time complexity analysis**
>
> We analyzed the practical complexity of LRPE in Section 4.3 and Table 9 in Appendix D and compared it with other methods. Here, we show our results again:
>
> | Method | Linear | Vanilla |
> | ------ | ------ | ------- |
> | Base   | 1      | 1       |
> | RoPE   | 0.95   | 0.97    |
> | SPE    | 0.42   | 0.41    |
> | PER    | 0.88   | -       |
> | T5     | -      | 0.7     |
> | LRPE   | 0.91   | 0.95    |
>
> In general, LRPE does not incur a significant computational burden to the transformer and can fulfill the practical needs by maintaining comparable efficiency.

---

> ### Comment · Reviewer_YduU · 2022-11-28
> **thanks for the answer**
>
> Thank you for your answer and the further experiments, which are interesting. I maintain my score, because I think the paper is definitely nice and a good read, and brings an incremental contribution.

---

### Official Review · Reviewer_o67a · 2022-10-25

**Confidence:** 5
**Correctness:** 4
**Technical Novelty And Significance:** 3
**Empirical Novelty And Significance:** 3
**Recommendation:** 6

**Clarity, Quality, Novelty And Reproducibility:**

**Clarity**
The paper is very well written, however several parts can be simplified (see below my comments)

**Quality**
Overall setup of experiments, math and ablations looks strong. The issue mainly with results which I specified above and below. At least better justification is needed for machine translation.

**Novelty**
The work is definitely new and provide extension on top of many embedding by unification of their formulation. Also it heavily extends RoFormer mainly adding additional projection matrix and different variations of orthogonal transformations.

**Reproducibility**
Details are extensively given in the paper with fairseq code link and hyperparameters in the Appendix.

**Major comments and suggestions to improve the paper**
- introduction: mentioning about RoFormer - first it is multiplicative embedding which is very different from a huge amount of embeddings we used before (it was only additive) and also it is general enough formulation, not just individual work I think. I think more careful formulation should be done in the text about RoFormer.
- Sec 2 background on the relative positional embedding. I think authors should mention that in the end relative positional embedding is not so efficient: it gives extra computational cost and memory, which slow down training (e.g. in LM/MT domains people mostly use only absolute positional embedding anyway, in speech it is observed 2-3x slow down, see [3]). Also there is generalization issue with relative positions still, see [3]. As a solution people developed things like [1-4] in the past year. Also what do authors mean by "more flexible"? In that sense sinusoidal pos. embedding is good, it is applicable to any position. Also worth to clearly mention that a lot of works tried to modify attention mechanism directly to introduce positional embedding which is not generalizable to linear transformers.
- Eq 6 - introduce $m$ here, otherwise not clear what is it.
- Eq 7 seems wrong to me as dot product is a value while $W_{t-s}$ is a matrix if compare Eq 7 and 8.
- I think RoFormer is the closest work to the current paper in the sense of unitary operator formulation and multiplicative nature of embedding. That is why I prefer to see more detailed discussion of RoFormer and formulation in 3.1.1 RoFormer in terms of the proposed framework, showing what is the main difference and last step (e.g. additional projection matrix on top of rotational matrix) RoFormer did not do and why more general formulation performs better in experiments.
- Sec 3.1.2 - usage of tilda could be reduced, just use notation of hat.
- Definition 3.1 Why we assume that matrices $M_t$ and $M_s$ the same? We could say that $W_{t-s}$ should decompose into $M_t M'_s$ - more general formulation.
- Be consistent on usage words encoding <-> embedding
- LRPE and URPE, its equivalence and Proof of Theorem 3.3 can be simplified and text could be squeezed. It is in the end a bit trivial. I have several suggestion how to improve this. First do not introduce both LRPE and URPE, but just LRPE and then with several rows only show that this leads to $W_{t-s}$ being unitary and equal to $W_t W_s$. To show this add after definition 3.1:
  - $W_0=I_d = M_s^H M_s$ -> $M_s$ is unitary. $W$ is decomposed as product of unitary matrices, thus W is also unitary (as unitary matrices is a group, or check by definition).
  - Then for $s=0$ we have $W_t=M_0^H M_t$ and then multiply by identity which is equal to $M_0^H M_0$ and get $W_{t-s}=M_s^H M_t=M_s^H (M_0 M_0^H) M_t= (M_0^H M_s)^H (M_0^H M_t) = W_s^H W_t$.
  - Appendix C3 can be removed then entirely. Also theorem C1 and C2 are about Jordan form for unitary matrices, so this can be simplified as reference.
  - Right now it is very confusing and hard to read. With above simplification it should be very obvious what is happening and no need for two definition. We just want to have some decomposition, and from it and our reasonable assumption on $W_0=I_d$ we get particular form on matrices and unitary property.
- Sec 3.3.2, please add what is $P$ matrix: projection, arbitrary? later it is clear, but better to introduce right away.
- From section 3.3.2 it is clear that mostly new formulation adds only some linear transformation on top of the rotational matrix. I think it is in line with work Li, Y., Si, S., Li, G., Hsieh, C.J. and Bengio, S., 2021. Learnable fourier features for multi-dimensional spatial positional encoding. Advances in Neural Information Processing Systems, 34, pp.15816-15829. where it was shown that additional transformation on top of the positional embedding is very helpful. I think it is worth to quantify this difference in the paper and perform ablation where matrix $P$ is selected as identity.
- Table 2: what about Transfromer-XL baseline?
- Could author give more info on why having relative positional embedding in the decoder for linear attention doesn't allow models to converge? is it true also for the baselines or only for proposed rel.pos.?
- Table 4: please mark with bolt only valid performance, as we select best based on valid and then look at the test. It is misleading just mark with bolt test on its own. Then we clearly will see that for linear attention all proposed rel.pos. actually worse than the baseline and for vanilla one almost same story (there is huge variation on test for two embeddings performing same on valid, 1BLEU difference, so more seeds or experiments are needed here).
- Eq 23 Appendix - values are forgotten in the formula
- could authors confirm that for all experiments the same model is used and only positional embedding is varied? Maybe it is worth to clearly state in the text (maybe I missed this).

**Missing citations:**
- [1] KERPLE: Kernelized Relative Positional Embedding for Length Extrapolation https://arxiv.org/abs/2205.09921 NeurIPS 2022
- [2] (AliBi) Ofir Press, Noah Smith, and Mike Lewis. Train short, test long: Attention with linear biases enables input length extrapolation. In International Conference on Learning Representations, 2022.
- [3] (CAPE) Likhomanenko, T., Xu, Q., Synnaeve, G., Collobert, R. and Rogozhnikov, A., 2021. CAPE: Encoding relative positions with continuous augmented positional embeddings. Advances in Neural Information Processing Systems, 34, pp.16079-16092.
- [4] (SHAPE) Kiyono, S., Kobayashi, S., Suzuki, J. and Inui, K., 2021, November. SHAPE: Shifted Absolute Position Embedding for Transformers. In Proceedings of the 2021 Conference on Empirical Methods in Natural Language Processing (pp. 3309-3321).
- [5] Dai, Z., Yang, Z., Yang, Y., Carbonell, J.G., Le, Q. and Salakhutdinov, R., 2019, July. Transformer-XL: Attentive Language Models beyond a Fixed-Length Context. In Proceedings of the 57th Annual Meeting of the Association for Computational Linguistics (pp. 2978-2988).



**Details Of Ethics Concerns:**

No concerns as work is about general formulation of positional embedding for transformer architecture.

**Strength And Weaknesses:**

**Strength**
- general formulation of relative positional embedding for linear attention which can be used now for both linear and vanilla transformers (many proposed embedding can be rewritten in terms of the proposed general formulation)
- nice simple math & extension of RoFormer
- experiments with many variations of proposed general relative positional embedding for language models, machine translation and downstream NLP tasks.

**Weaknesses**
- Paper can be simplified to have only LRPE definition and inheriting the proper form with assumption $W_0=I_d$
- Ablation on usage $P$ as identity to understand the effect of the extra linear transformation on top of the rotational operation. Why do we need $P$ matrix at all as in examples it is still unitary / orthogonal, why this decomposition into $P$ and $\Lambda$ is helpful? There is one ablation in Table 5, but would be great to have more and see consistent picture here. Any analysis on understanding why we need $P$ would be very useful. What will happen if we learn this matrix for baselines too?
- Results in Table 3 for vanilla transformer are not convincing. Seems we even don't need any relative information to perform the best.
- Machine Translation experiments are weak. It is known that test and validation for WMT-14 En-De are not very well correlated, probably due to small data size. Here all results should be reported with statistical significance, otherwise it is too huge variation could be. I would suggest to redo experiments either with several seeds or for En-Fr which is larger and stable results are observed there. Improvements 0.05 authors reports here is out of statistical significance, so it should be ignored for now.
- I think overall experimental outcomes shows that there is huge variability between embeddings and moreover there is no one formulation which behaves consistently better in all experiments over the baselines. In this case I am not sure how general formulation can be helpful from practical point. Yes, theoretical formulation and math behind is very cool, but it could be the case that it is not practical in the end and does not bring too much improvement (some negative result that original baselines are good enough). Maybe more broader experiments in other domains, like vision and speech could help to resolve this issue, but it is future work obviously.

**Summary Of The Paper:**

Positional embedding is an important component of transformer to give position information and also provide better generalization and usage of longer context. Recently several papers attempted to resolve the problem of generalization to longer sequences as well as better designing relative positional embedding. The main issue was applicability to popular optimized linear attention as relative positional embedding is modifying attention and was not applicable. RoFormer was the first to formulate properly the problem with unitary transformation (by rotating on some angle) and propose multiplicative embedding to be applicable to linear attention. This paper continue this generalization: it formulates general property we need for linear attention from positional embedding and obtains generalization of RoFormer with unitary transformation and additional projection matrix. With that many relative positional embeddings can be reformulated in the proposed unification, many new designs are further proposed in the paper with different unitary and projection matrices. All of them are tested for language modeling, machine translation and downstream NLP tasks. In many cases different proposed embeddings improve results over the baselines and previously proposed relative positional embeddings.

**Summary Of The Review:**

The paper proposes general formulation of relative positional embedding which is applicable to linear and vanilla transformer due to multiplicative nature. I like a lot math extension and general formulation of RoFormer with unitary operators which is done in the paper. Finally we have very fundamental (and correct from math point) definition of relative positional embedding (done with RoFormer too, but it is extended further). The paper gives overview of many embeddings which can be formulated in terms of the proposed definition. Authors study different variations of matrices which are used in the decomposition of the general relative positional embedding. However, from experiments on language modeling, machine translation and downstream performance on NPL tasks we can see that there is no one positional embedding from proposed formulation which performs the best among others in all tasks. Also there are weak experiments and justification in machine translation domain and NLP downstream tasks. Moreover I think the main key point of having some projection matrix is not studied well and its effect is not understood.

**Update** After rebuttal and discussion: With additional experiments and ablations authors made the paper stronger on both technical and empirical sides. I change correctness evaluation from 2 to 4. I support acceptance of the paper, but keeping score of 6 as empirical results are not showing consistent improvement for one proposed embedding and there is no score of 7 :) . In this light I suggest to rephrase a bit the text to make it clear that the focus of the paper is the general framework itself and the theory under it, not one new embedding in particular to perform the best. Also I encourage authors to include all results from rebuttal period, including vision domain with ViT.

---

> ### Author Response · Authors · 2022-11-18
> **Response to Reviewer o67a(Part 1)**
>
> **Q1. Paper can be simplified to have only LRPE definition and inheriting the proper form with assumption $\mathbf W_0=\mathbf I_d$.**
>
> Thanks for your suggestion, we will revise the paper accordingly in the revised version.
>
> **Q2. Why do we need $\mathbf P$ matrix at all as in examples it is still unitary/orthogonal, why this decomposition into $\mathbf P$ and $\mathbf \Lambda_t$ is helpful? More Ablation on $\mathbf P$.**
>
> It is worth noting that we do not deliberately decompose the matrix $\mathbf W_t$  into the form of  $\mathbf P^{\mathbf H} \mathbf \Lambda_t \mathbf P$. It is derived from Eq. 14: $\mathbf W_t$  is a unitary matrix and according to the “spectral decomposition of a unitary matrix", we can derive it into the form of $\mathbf P^{\mathbf H} \mathbf \Lambda_t \mathbf P$. In this form, we can easily assemble multiple variants of RPE and we list 8 examples in the manuscript. The purpose of Table 5 is to analyze whether different $\mathbf P$ will affect the results. We add more results in Table 5 in the revised version.
>
> **Q3. What will happen if we learn this unitary matrix $\mathbf P$ for baselines too?**
>
> For the baseline, the unitary matrix $\mathbf P$ will be canceled out as follows:
>
> $$
> \begin{equation}
> \left(\mathbf{P}\mathbf{q}_s\right)^{\mathrm{H}}\left(\mathbf{P} \mathbf{k}_t\right)
> =\mathbf{q}_s^{\mathrm{H}} \mathbf{P}^{\mathrm{H}} \mathbf{P} \mathbf{k}_t=
> \mathbf{q}_s^{\mathrm{H}}  \mathbf{k}_t
> \end{equation}
> $$
>
> **Q4. The results in Table 3 for the vanilla transformer are not convincing. Seems we even don't need any relative information to perform the best.**
>
> The reasons for this are twofold:
>
> - The pre-trained corpus is not large enough compared to finetune corpus, which leads to over-fitting during pre-training. From Table 8, in the pre-training stage, most Rpe can reduce the validation loss.
> - We use the same hyperparameters for all methods for fair comparison (refer to the last paragraph of 4.2 Bidirectional language model). In fact, the optimal hyperparameters for each method may not be exactly the same.
>
> **Q5. Machine Translation experiments Results**
>
> The results in Table 4 represent the averaged results of five separate trials so that the results are statistical significance. We list part of the detailed results below.
>
> Linear attention:
>
> | method | loss  | bleu(valid) | bleu(test) |
> | ------ | ----- | ----------- | ---------- |
> | Type4  | 4.105 | 29.84       | 26.72      |
> |        | 4.101 | 29.81       | 27.4       |
> |        | 4.11  | 29.72       | 27.61      |
> |        | 4.103 | 29.74       | 27.9       |
> |        | 4.1   | 29.8        | 27.7       |
> | Type7  | 4.103 | 29.84       | 27.34      |
> |        | 4.101 | 29.81       | 27.37      |
> |        | 4.108 | 29.83       | 27.43      |
> |        | 4.098 | 29.86       | 27.46      |
> |        | 4.094 | 29.77       | 27.64      |

---

> > ### Comment · Reviewer_o67a · 2022-11-25
> > **Discussion**
> >
> > Dear authors,
> >
> > Thanks for comments and new revision. I went through the revision, looks much simpler and readable with simplification on URPE -> LRPE.
> >
> > Some comments and extra questions:
> >
> > Q3: Yep, it will cancel, but as we know this reparametrization can influence the optimization and P will affect the gradient computation in some sense. So I wonder if it will make any changes from optimization perspective.
> >
> > Q4: Yep, totally agree and I believe that your setup of experiments is the best (that you used the same hyper parameters). My point is more that maybe another data will be more convincing - it is ok that for some data we don't need to have RPE.
> >
> > Q5: Thanks for the updated Table 4 with std.
> >
> > Q6: Thanks for extra experiments, looks very strong. Could you specify what ViT recipe did you use to train? What positional embedding is used in the baselines?
> >
> > Q8: Yep, not simple math then. But wonder if we still can get something general.
> >
> > Q9: Did you rerun Transformer-XL or the numbers are from the prior work? It is a bit strange that XL gives way worse results even than the baseline with sinusoidal positional embedding.
> >
> > Q10: this is really very interesting. I would suggest to highlight this in the paper for future works to cite properly your work.

---

> > > ### Author Response · Authors · 2022-12-01
> > > **Response to Reviewer o67a(Part 3)**
> > >
> > > **Q3: Yep, it will cancel, but as we know this reparametrization can influence the optimization and P will affect the gradient computation in some sense. So I wonder if it will make any changes from an optimization perspective.**
> > >
> > > Sure, the reparametrization can sometimes affect the optimization. However, in our case, we can prove that in theory, the gradients are also independent of $\mathbf P$, i.e., $\mathbf P$ will not influence the optimization. We provide the proof below.
> > >
> > > Let us analyze the backward pass. We use the following notations:
> > >
> > > $$
> > > \begin{aligned}
> > > \mathbf Q, \mathbf K, \mathbf V \in \mathbb C^{n\times d}\newline
> > > \mathbf P \in \mathbb C^{d\times d}, \mathbf P^{\mathbf H}\mathbf P=\mathbf I_d ,
> > > \mathbf Q_1 =\mathbf Q \mathbf P\in \mathbb C^{n\times d}, \mathbf K_1 =\mathbf K \mathbf P\in \mathbb C^{n\times d}\newline
> > > \mathbf A=\mathbf Q_1  \mathbf K_1^{\mathbf H}\in \mathbb C^{n\times d}
> > > \end{aligned}
> > > $$
> > >
> > > and
> > >
> > > $$
> > > \begin{aligned}
> > > {[\nabla_{\mathbf{M}} \mathcal{L}]}\_{ij}=\frac{\partial \mathcal{L}}{\partial m_{i j}}
> > > \end{aligned}
> > > $$
> > >
> > > where $\mathcal L$ stands for the loss function, and $\mathbf M$ is a parameter matrix.
> > >
> > > Given $\nabla_{\mathbf{A}} \mathcal{L}\in \mathbb R^{n\times n}$, we have:
> > >
> > > $$
> > > \begin{aligned}
> > > \nabla_{\mathbf{Q_1}} \mathcal{L}&=(\nabla_{\mathbf{A}} \mathcal L)\mathbf K_1 \newline
> > > \nabla_{\mathbf{Q}} \mathcal{L}&=(\nabla_{\mathbf{Q_1}} \mathcal L)\mathbf P^{\mathbf H} \newline
> > > &= (\nabla_{\mathbf{A}} \mathcal L)\mathbf K_1 \mathbf P^{\mathbf H}  \newline
> > > &=(\nabla_{\mathbf{A}} \mathcal L)\mathbf K \mathbf P \mathbf P^{\mathbf H} \newline
> > > &= (\nabla_{\mathbf{A}} \mathcal L)\mathbf K \newline
> > > \nabla_{\mathbf{K_1}} \mathcal{L}&=(\nabla_{\mathbf{A}} \mathcal L)\mathbf Q_1  \newline
> > > \nabla_{\mathbf{K}} \mathcal{L}&=(\nabla_{\mathbf{K_1}} \mathcal L)\mathbf P^{\mathbf H} \newline
> > > &= (\nabla_{\mathbf{A}} \mathcal L)\mathbf Q_1 \mathbf P^{\mathbf H} \newline
> > > &=(\nabla_{\mathbf{A}} \mathcal L)\mathbf Q \mathbf P \mathbf P^{\mathbf H} \newline
> > > &= (\nabla_{\mathbf{A}} \mathcal L)\mathbf Q .
> > > \end{aligned}
> > > $$
> > >
> > > As shown, the gradient term is independent of $\mathbf P$.
> > >
> > > **Q4: Yep, totally agree and I believe that your setup of experiments is the best (that you used the same hyperparameters). My point is more that maybe other data will be more convincing - it is ok that for some data we don't need to have RPE.**
> > >
> > > Thanks for your suggestion, we are testing LRPE on other tasks and benchmarks and will include them in the revised version.
> > >
> > > **Q6: Thanks for the extra experiments, looks very strong. Could you specify what ViT recipe did you use to train? What positional embedding is used in the baselines?**
> > >
> > > We choose DeiT-small as our baseline and use a learnable positional embedding. For the model with LRPE, we use Type 2 and operate in $h$ and $w$ dimensions respectively, where $h$ and $w$ are the length and width dimensions of the feature.
> > >
> > > **Q8: Yep, not simple math then. But wonder if we still can get something general.**
> > >
> > > Sure, we provide a preliminary analysis of the general case below. Given a general condition:
> > >
> > > \begin{aligned}
> > > f_{\mathrm{rel}}\left(\mathbf{q}_s, \mathbf{k}_t\right)
> > > &=\mathbf{q}_s^{\mathrm{H}} \mathbf{W}_m \mathbf{k}_t \newline
> > > &=\left(\mathbf{M}_s \mathbf{q}_s\right)^{\mathrm{H}} \left(\mathbf{N}_t \mathbf{k}_t\right) \newline
> > > &=\mathbf{q}_s^{\mathrm{H}} \mathbf{M}_s^{\mathrm{H}} \mathbf{N}_t \mathbf{k}_t, \newline
> > > m&=t-s \newline
> > > \mathbf W_0 &=\mathbf I_d .
> > > \end{aligned}
> > >
> > > We can simplify it to:
> > >
> > > $$
> > > \mathbf{W}_{t-s} =\mathbf{M}_s^{\mathrm{H}} \mathbf{N}_t
> > > $$
> > >
> > > Take $t=s$, we get:
> > >
> > > $$
> > > \mathbf{M}_t^{\mathrm{H}} \mathbf{N}_t=\mathbf W_0 =\mathbf I_d,\\
> > >  \mathbf M_t = (\mathbf N_t^{-1})^{\mathbf H} .
> > > $$
> > >
> > > Then:
> > >
> > > $$
> > > \mathbf W_{t-s}=(\mathbf N_s^{-1})^{\mathbf H} \mathbf N_t.
> > > $$
> > >
> > > Take $s=0$, we get:
> > >
> > > $$
> > > \mathbf W_{t}=(\mathbf N_0^{-1})^{\mathbf H} \mathbf N_t.
> > > $$
> > >
> > > Replace $t$ with $t-s$, we get:
> > >
> > > $$
> > > \mathbf W_{t-s}=(\mathbf N_0^{-1})^{\mathbf H} \mathbf N_{t-s}= (\mathbf N_s^{-1})^{\mathbf H} \mathbf N_t.
> > > $$
> > >
> > > Finally, the problem is equivalent to:
> > >
> > > $$
> > > (\mathbf N_0^{-1})^{\mathbf H} \mathbf N_{t-s}= (\mathbf N_s^{-1})^{\mathbf H} \mathbf N_t.
> > > $$
> > >
> > > The problem is difficult to solve. A feasible solution could be to use the Jordan canonical form of the matrix. We will study this problem later.

---

> > > > ### Comment · Reviewer_o67a · 2022-12-06
> > > > **Final score**
> > > >
> > > > Dear authors,
> > > >
> > > > Thanks for productive discussion and for latest explanations. On Transformer-XL - agree, and I believe original paper had a bit different recipe. I have updated my summary on the recommendation. Doubling it here for visibility:
> > > >
> > > > With additional experiments and ablations authors made the paper stronger on both technical and empirical sides. I change correctness evaluation from 2 to 4. I support acceptance of the paper, but keeping score of 6 as empirical results are not showing consistent improvement for one proposed embedding and there is no score of 7 :) . In this light I suggest to rephrase a bit the text to make it clear that the focus of the paper is the general framework itself and the theory under it, not one new embedding in particular to perform the best. Also I encourage authors to include all results from rebuttal period, including vision domain with ViT.
> > > >
> > > > Best, Reviewer.

---

> > > > > ### Author Response · Authors · 2022-12-09
> > > > > **Response to Reviewer o67a**
> > > > >
> > > > > Dear Reviewer o67a,
> > > > >
> > > > > Thanks for your insightful suggestion. We will revise the manuscript accordingly.
> > > > >
> > > > > Kind regards,
> > > > > Authors.

---

> > > ### Author Response · Authors · 2022-12-01
> > > **Response to Reviewer o67a(Part 4)**
> > >
> > > **Q9: Did you rerun Transformer-XL or the numbers are from the prior work? It is a bit strange that XL gives way worse results even than the baseline with sinusoidal positional embedding.**
> > >
> > > Yes, we re-ran Transformer-XL using [Fairseq](https://github.com/facebookresearch/fairseq/blob/main/examples/adaptive_span/README.md) implementation. There are two possible reasons for this phenomenon:
> > >
> > > - use the same hyperparameters as our other methods;
> > > - in order to be consistent with the previous model, Adaptive Softmax was not used;
> > >
> > > We list our model configuration below:
> > >
> > > ```python
> > > @register_model_architecture("transformer_xl", "transformer_xl_base")
> > > def transformer_xl_base(args):
> > >     base_lm_architecture(args)
> > >     args.cutoffs = [260000]
> > >     args.d_model = 512
> > >     args.n_head = 8
> > >     args.d_head = args.d_model // args.n_head
> > >     args.d_inner = 2048
> > >     args.div_val = 1
> > >     args.n_layer = 6
> > >     args.mem_len = 128
> > >     args.clamp_len = -1
> > >     args.same_length = False
> > >     args.dropout = 0.0
> > >     args.dropatt = 0.0
> > > ```
> > >
> > > **Q10: this is really very interesting. I would suggest to highlight this in the paper for future works to cite properly your work.**
> > >
> > > Thanks for the suggestion, we will include this in our future work and highlight it in the revised version.

---

> ### Author Response · Authors · 2022-11-18
> **Response to Reviewer o67a(Part 2)**
>
> **Q6. The value of the general formula and more experiments**
>
> Thanks for your suggestion. The main contribution of this paper is to provide a general form of Rpe for linear attention and present a systematic study of the conditions, i.e., Eq. 14, which has not been explored in previous papers. In our general form of Rpe, some existing methods such as Rope and PermutateFormer can be derived as well as some new solutions such as solution 1, solution 2, and solution 3. It is hard to find an optimal Rpe that performs well for all tasks, but we provide a good starting point. Researchers can derive new Rpes for the target tasks for better performance, i.e., choosing a different orthogonal matrix $\mathbf P$.
>
> As suggested, we extend our method to the image classification task. Specifically, we conducted image classification on CIFAR-100 and ImageNet1K. We adopted the ViT network structure and report the results of both vanilla attention and linear attention with and without LRPE in the table below. All models are trained from scratch.
>
> “lvit" means model variants that replace the attention modules of ViT with linear attention (1+elu), “+lrpe" denotes model variants equipped with LRPE.
>
> CIFAR-100：
>
> | method    | lr     | batch size | acc   | params |
> | --------- | ------ | ---------- | ----- | ------ |
> | lvit      | 0.0005 | 200        | 64.57 | 22m    |
> | lvit+lrpe | 0.0005 | 200        | 68.95 | 22m    |
> | vit       | 0.0005 | 200        | 71.85 | 22m    |
> | vit+lrpe  | 0.0005 | 200        | 74.20 | 22m    |
>
> ImageNet-1k：
>
> | method    | lr     | batch size | acc   | params |
> | --------- | ------ | ---------- | ----- | ------ |
> | lvit      | 0.0005 | 1600       | 76.87 | 22m    |
> | lvit+lrpe | 0.0005 | 1600       | 79.15 | 22m    |
> | vit       | 0.0005 | 1600       | 80.22 | 22m    |
> | vit+lrpe  | 0.0005 | 1600       | 80.83 | 22m    |
>
> As shown, our LRPE brings significant improvements in both linear attention and vanilla attention. In CIFAR-100, it improves the Top 1 accuracy by 4.38 on linear attention and 2.35 on vanilla attention. In ImageNet, it also improves Top1 accuracy by 2.28 and 0.61 respectively. We will include these experiments in the revised paper.
>
>
>
> For suggestions to improve the paper, we directly update the paper accordingly in the revised version. Below we answer part of the questions:
>
> **Q7. Meaning of “more flexible”.**
>
> The “more flexible” means that we can get many types of Rpe based on Eq 14.
>
> **Q8. Definition 3.1 Why do we assume that matrices $\mathbf M_t$ and $\mathbf M_s$ are the same? We could say that $\mathbf W_{t-s}$ should decompose into $\mathbf M_t^{\mathbf H} \mathbf M_s'$ , a more general formulation.**
>
> We made this assumption here for better mathematical handling. In the more general formulation, the following derivation would be hard to process.
>
> **Q9. Transfromer-XL baseline**
>
> We have added the results of Transformer-XL in Table 2.
>
> **Q10. Could the author give more info on why having relative positional encoding in the decoder for linear attention doesn't allow models to converge? Is it true also for the baselines or only for proposed rel.pos.?**
>
> We empirically found that using linear attention in nmt's decoder will prevent models from converging regardless of whether Rpe is used or not, even for the baseline model. We think this is an intriguing phenomenon and will incorporate it into our future work.
>
> **Q11. Could authors confirm that for all experiments the same model is used and only positional encoding is varied? Maybe it is worth clearly stating in the text (maybe I missed this).**
>
> Yes, the same model is shared in all experiments, training hyperparameters, such as batch size, update times, etc., the only difference is the type of Rpe. We mention this in the Training configuration part of 4.1 and the detailed configuration is in table 7.

---

### Official Review · Reviewer_JvG7 · 2022-10-26

**Confidence:** 3
**Correctness:** 3
**Technical Novelty And Significance:** 3
**Empirical Novelty And Significance:** 3
**Recommendation:** 6

**Clarity, Quality, Novelty And Reproducibility:**

Clarity
- this work, is well written, can be easily understood for readers with strong background in the formalization of vanilla transformer, linear transformers.

Quality
- the work is well organized, starting from a motivation for why a new variant of URPE, relation with linear transformation, and formalization and experimental sections.

Novelty
- achieving a complexity of O(n) for PE, this work, adds a novel contribution to linear transformers. Moreover, the approach is flexible enough to be applied in the standard transformer.

Reproducibility
- requires multiple pass to understand the theoretical background
- the provided pseudo-code of the URPE can be utilized to replicate the approach.

**Strength And Weaknesses:**

Strengths
- Unlike recurrent networks, transformers are dependent on the PE for learning the sequential arrangement of the input representation. In the past, literature relied either on an absolute, learned/relative PE's. As an integral part of transformer, this work, provides a principled approach to simplify PE for linear computation.
- In addition to a comparable result in a LM and MT tasks, the proposed URPE approaches preserves a linear (O(n)) compute complexity, with a relatively minor (9%) training speed delay.

Weaknesses
-  Despite the results for tasks the LM, MT, and text classification, the experimental setting is quite narrow. Given the depth of formalization of both the vanilla, linearized and the proposed URPE approaches, this work, can be benefited from more tasks and broad experimental settings. For instance, summarization, given longer sequence length input and document level MT can be a good starting point to further probe the complexity vs quality metrics.
- Despite 8 variants of URPE (Table 1), this work, lacks a detailed comparison and analysis of each variants. Only performance difference for LM and MT tasks are provided.

**Summary Of The Paper:**

This work proposes new variants of relative positional encoding (PE) that are applicable for linear transformers - relative PE with unitary transformation (URPE). The proposed URPE variants preserve a linear time-space complexity, and demonstrates a comparable performance against the standard/vanilla transformer, across tasks such as LM and MT.

**Summary Of The Review:**

This work, proposes a new variant of relative positional encoding, that can achieve a complexity of O(n). The proposed approach can be applied both for linear and the vanilla transformers. The proposed approach, with a detailed theoretical background, shows comparable result with strong baseline and related work. Although, the presented result can be benefit from additional tasks such as summarization.

---

> ### Author Response · Authors · 2022-11-18
> **Response to Reviewer JvG7**
>
> **Q1. More experiments on different tasks such as summarization.**
>
> As suggested, we additionally test our model on the summarization task on the CNN-Dailymail dataset and the image classification task on the CIFAR-100 and ImageNet-1K datasets.
>
> For the summarization task, we use the Bart-base model. All models are trained from scratch and updated 20k times under the same configurations.
>
> We show the results of the CNN-Dailymail dataset below.
>
> Linear Attention:
>
> | method      | loss(valid) |
> | ----------- | ----------- |
> | Base        | 7.287       |
> | Rope        | 7.032       |
> | SPE         | 7.296       |
> | Householder |             |
> | Type1       | 7.077       |
> | Type2       | 6.936       |
> | Type3       | 6.913       |
> | Permutation |             |
> | Type5       | 7.04        |
> | Type6       | 6.894       |
>
> Vanilla Attention:
>
> | method      | loss(valid) |
> | ----------- | ----------- |
> | Base        | 7.242       |
> | Rope        | 5.901       |
> | SPE         | 7.27        |
> | T5          | 7.248       |
> | Householder |             |
> | Type1       | 5.913       |
> | Type2       | 5.944       |
> | Type3       | 5.944       |
> | Type4       | 6.113       |
> | Permutation |             |
> | Type5       | 5.919       |
> | Type6       | 5.934       |
>
> Our method achieves lower validation loss than the base model and most previous methods.
>
> We show the results of the image classification tasks in R2Q7.
>
> **Q2. Analysis of LRPE**
>
> The main contribution of this paper is to provide a unified solution for encoding relative position information to linear transformers. To systematically discuss the solution and illustrate the use of the sufficient condition for adaptation, i.e., Eq. 14, we provide 3 families of solutions. In fact, we can form infinite numbers of solutions based on Eq. 14 as the $\mathbf P$ is any unitary matrix. We present 8 examples to instantiate the three families of solutions and compare their performance on different tasks. The detailed comparison of various solutions is not the primary focus of this paper. In fact, some of these relative positional encoding methods may be orthogonal to others, the only connection between them is that they all can be unified as  $\mathbf P^{\mathbf H} \mathbf \Lambda_s \mathbf P$.

---

### Author Response · Authors · 2022-11-18
**About the revision of the paper.**

Thanks to the suggestions of the Reviewers, we have revised the paper, the main changes are as follows:
- Add more discussion about RoPE in sections 1 and 2,;
- Simplify the proof part, delete the definition of URPE, and only keep LRPE;
- Change method name to LRPE:
- Add transformer-xl results in Table 2;
- Table 4 is updated with the results of the average of 5 experiments;

---

### Decision · Program_Chairs · 2023-01-20

**Decision:**

Reject

**Justification For Why Not Higher Score:**

Various concerns have raised during the reviewing process, including the novelty, weak experimental performance and limited evaluation tasks. The authors have not fully addressed these concerns.

Also, even during rebuttal, results on additional tasks are reported, considering the motivation of the work, the proposed approach should be tested on NLP tasks with long inputs. e.g., various datasets proposed on long narrative stories such as books and movie scripts.

**Justification For Why Not Lower Score:**

N/A

**Metareview: Summary, Strengths And Weaknesses:**

Various concerns have raised during the reviewing process, including the novelty, weak experimental performance and limited evaluation tasks. The authors have not fully addressed these concerns.

Also, even during rebuttal, results on additional tasks are reported, considering the motivation of the work, the proposed approach should be tested on NLP tasks with long inputs. e.g., various datasets proposed on long narrative stories such as books and movie scripts.